# 🦜 Parrot: Multilingual Visual Instruction Tuning

**Hai-Long Sun** [1 2 3 *]  **Da-Wei Zhou** [1 2]  **Yang Li** [3]  **Shiyin Lu** [3]  **Chao Yi** [1 2]  **Qing-Guo Chen** [3]  **Zhao Xu** [3]
**Weihua Luo** [3]  **Kaifu Zhang** [3]  **De-Chuan Zhan** [1 2]  **Han-Jia Ye** [1 2]

## Abstract

The rapid development of Multimodal Large Language Models (MLLMs), such as GPT-4o, marks a significant step toward artificial general intelligence. Existing methods typically align vision encoders with LLMs via supervised fine-tuning (SFT), but this often deteriorates their ability to handle *multiple languages* as training progresses. We empirically observe that imbalanced SFT datasets, largely English-centric, degrade performance on non-English languages due to the failure in multilingual token alignment. To address this, we propose PARROT, a novel approach that leverages textual guidance for visual token alignment at the language level. PARROT conditions visual tokens on diverse language inputs and uses Mixture-of-Experts (MoE) to align multilingual tokens. By computing cross-attention between initial visual features and textual embeddings, we select the most relevant experts, converting visual tokens into language-specific representations. Additionally, we introduce the Massive Multilingual Multimodal Benchmark (MMMB), a new benchmark comprising 6 languages, 15 categories, and 12,000 questions, to assess multilingual capabilities. PARROT achieves state-of-the-art performance on both the multilingual benchmarks and a wide range of multimodal tasks. Code and dataset are available at: https://github.com/AIDC-AI/Parrot.

## 1. Introduction

The rapid development of Large Language Models (LLMs), such as GPT-4 (Radford et al., 2018; Brown et al., 2020;

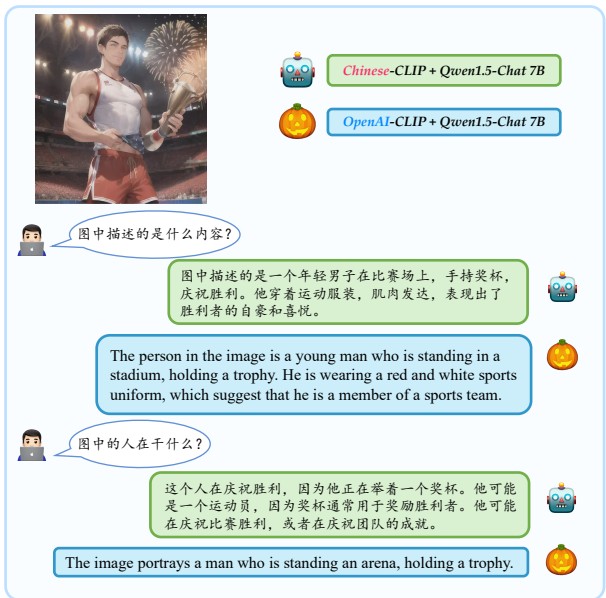

*Figure 1.* The output of OpenAI-CLIP-based and Chinese-CLIP-based models using the same Chinese prompts. We can observe that the OpenAI-CLIP-based model exhibits confusion between Chinese and English responses.

OpenAI, 2023a; 2024), has gained significant attention. However, LLMs are limited to processing only text-only data. The integration of visual modalities has endowed LLMs with multimodal capabilities (Wang et al., 2024a; Liu et al., 2024), driving the emergence of Multimodal Large Language Models (MLLMs). These models combine pre-trained LLMs with vision encoders, bridging the modality gap by aligning visual features with language embeddings. Current research predominantly uses either Q-Former (Li et al., 2023b) or MLP projector (Liu et al., 2023b) to align vision encoders with LLMs, enabling models to process multimodal.

Multilingual capability is a crucial aspect of MLLMs, enabling them to generate responses in the same language as the input, accommodating linguistic diversity. This feature is vital for ensuring equitable access to technology across different regions and cultures (Chen et al., 2022; Hu et al., 2023). Many LLMs (Grattafiori et al., 2024; Yang et al., 2024; OpenAI, 2023b) exhibit multilingual capabilities,

*Work done during his internship at Alibaba Group [1] School of Artificial Intelligence, Nanjing University [2] National Key Laboratory for Novel Software Technology, Nanjing University [3] AI Business, Alibaba Group. Correspondence to: Han-Jia Ye <yehj@lamda.nju.edu.cn>.

*Proceedings of the 42nd International Conference on Machine Learning*, Vancouver, Canada. PMLR 267, 2025. Copyright 2025 by the author(s).

generating diverse language responses based on prompts. However, after multimodal alignment training, MLLMs often lose their ability to effectively understand, process, or generate non-English languages. This phenomenon, which we refer to as *multilingual erosion*, results in models like LLaVA (Liu et al., 2023b), which tend to respond predominantly in English, even when the input is in another language. Therefore, it is essential to address multilingual erosion for improving MLLM's multilingual abilities.

The primary cause of multilingual erosion is the imbalanced nature of multimodal alignment data, which is overwhelmingly English-centric. While models align visual and textual tokens well in English, their performance in other languages is suboptimal. Through empirical analysis, we observe that multilingual erosion stems from the misalignment between visual tokens and textual tokens in non-English languages. For example, in our ablation study using OpenAI-CLIP and Chinese-CLIP (Yang et al., 2022), we find that a model using OpenAI-CLIP struggles with Chinese inputs, while the Chinese-CLIP-equipped model effectively understands and generates Chinese responses. As shown in Figure 6, the t-SNE visualizations further demonstrate that the visual features of Chinese-CLIP-based LLaVA are more closely aligned with the Chinese prompts. Therefore, an intuitive question is: how to transform the visual features into language-specific embeddings to enhance the MLLM's multilingual capabilities.

Due to the scarcity of non-English multimodal data (*e.g.*, the lack of large-scale, high-quality image-text datasets), it is necessary to use as little multilingual data as possible to enhance the model's multilingual capabilities. To this end, we propose a novel method, PARROT, which uses textual guidance to align visual tokens at the language level. PARROT leverages a Mixture-of-Experts (MoE) module (Jacobs et al., 1991) to convert visual tokens into language-specific embeddings. Specifically, we first calculate the cross-attention between the class token of visual features and the text embeddings. The resulting features are then passed through the MoE router to activate a probability distribution for each language expert. Based on the input language, visual tokens biased towards English are transformed into language-specific embeddings using the appropriate expert. This approach not only enhances the multilingual capabilities of the MLLM but also bridges the multimodal gap effectively.

To address the scarcity of current multilingual benchmarks, we introduce a new benchmark encompassing six languages: English, Chinese, Portuguese, Arabic, Turkish, and Russian. This includes an extension of the MMBench dataset to six languages and a Massive Multilingual Multimodal Benchmark (**MMMB**) featuring 2,000 evaluation questions per language, totaling 12,000 questions. Through a semi-automatic construction process, we mitigate the po-

tential erros and noise. Extensive experiments validate the PARROT's state-of-the-art performance across two multilingual benchmarks, surpassing Qwen2-VL and LLaVA-OneVision in multiple languages. Additionally, we evaluate our model across a broad range of multimodal benchmarks (*e.g.*, MME (Fu et al., 2023) and ScienceQA (Lu et al., 2022), and SEED-Bench (Li et al., 2024a)), demonstrating its competitive performance in diverse tasks.

## 2. MMMB: A Massive Multilingual Multimodal Benchmark

In this section, we first outline the limitations of existing benchmarks and identify the key characteristics an ideal multilingual benchmark should exhibit. We then provide a detailed explanation of how to construct a new benchmark.

### 2.1. Limitations of Existing Benchmarks

There are several existing multilingual benchmarks (*e.g.*, Multi30K (Elliott et al., 2016), M3Exam (Zhang et al., 2024b), MMBench (Liu et al., 2023c), and LLaVA-Bench (Liu et al., 2023b; Hu et al., 2023)) for MLLMs, but they have notable limitations: **1) Outdated Benchmarks.** Multi30k is designed for image-text retrieval tasks, and its performance has nearly reached its upper bound due to relatively simple problems. **2) Non-Standardized Evaluations.** Benchmarks like LLaVA-Bench rely on evaluations using GPT-4. This dependence on GPT-4 as a de facto "Ground Truth" may hinder reproducibility. Additionally, LLaVA uses a deprecated version (GPT-4-0314), which introduces potential inconsistencies if other versions are used, leading to unfair comparisons. Moreover, M3Exam lacks consistent test samples across different languages, making it difficult to determine whether poor performance results from the difficulty of the problem or the model's limited multilingual capabilities. **3) Limited Languages.** MMBench and LLaVA-Bench are restricted to English and Chinese, limiting their ability to assess multilingual capabilities across a broader range of languages.

### 2.2. The Key Characteristics of an Ideal Benchmark

To more accurately evaluate the multilingual capabilities of MLLMs, an ideal benchmark should exhibit the following characteristics:

**1) Languages with Significant Differences.** The benchmark should cover a diverse range of language families, selecting languages that are distinct and non-repetitive. This ensures a comprehensive assessment of MLLMs' ability to adapt across linguistic variations.

**2) Problems with Medium Level of Difficulty.** The problems should not be overly challenging (*e.g.*, logical rea-

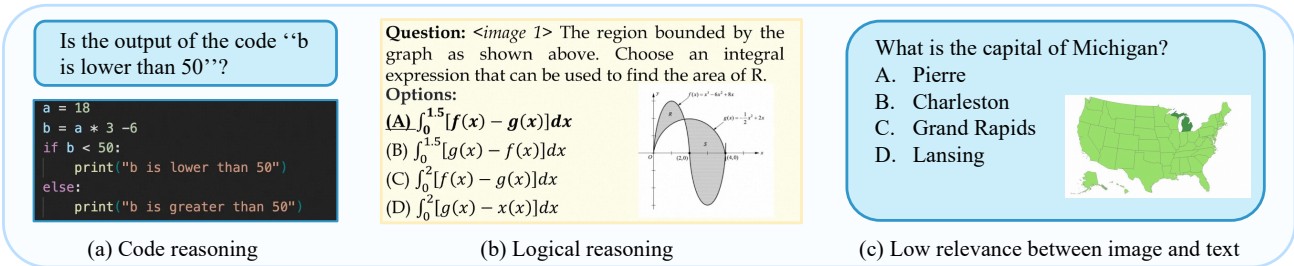

(a) Code reasoning      (b) Logical reasoning      (c) Low relevance between image and text

*Figure 2.* Some bad cases for the existing multilingual benchmark. **Left:** code reasoning is strongly related to English. **Middle:** logical reasoning is too challenging. **Right:** lack relevance between image and text.

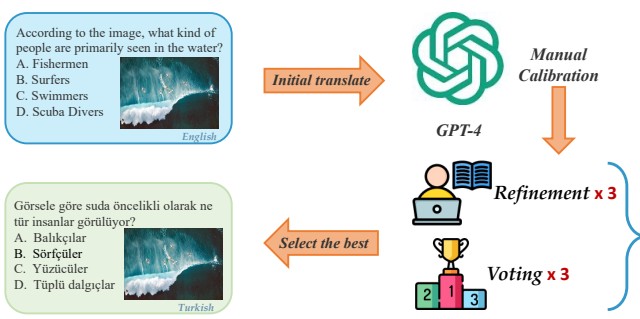

*Figure 3.* The calibration process for constructing a multilingual benchmark consists of two stages: translation and calibration.

soning), as the primary goal is to assess the multilingual understanding, processing, and generating capabilities of MLLMs, rather than their reasoning abilities.

**3) Tasks with Multilingual and Multimodal.** As shown in Figure 2, datasets should not be overly reliant on English (*e.g.*, code reasoning). These tasks should not inherently translate into multiple languages if they are composed mainly of English words. Moreover, images should be an indispensable part when MLLMs answer the question. For instance, when shown a map of the United States and asked to identify its capital, relying solely on text-based abilities would be insufficient. Therefore, it is crucial that questions exhibit a significant interplay between images and text.

**4) Content Consistency across Languages.** To fairly assess the multilingual capabilities of MLLMs, content across languages must remain consistent. For instance, if English questions focus primarily on *addition within one hundred*, while Chinese questions emphasize *calculus computation*, it would be difficult to determine whether poor performance in Chinese stems from the complexity of the problem or from the model's limited multilingual capabilities. Ensuring content consistency across languages is thus essential for a fair and accurate comparison.

### 2.3. Construction Pipeline

Following the above criteria, we select six languages for inclusion: English (*en*), Chinese (*zh*), Portuguese (*pt*), Arabic

(*ar*), Turkish (*tr*), and Russian (*ru*). These languages represent a diverse range of linguistic families. Detailed information and multilingual examples are provided in Figure 4. Regarding dataset requirements and consistency, our benchmark is constructed with two key considerations: **1)** Since MMBench (Liu et al., 2023c) officially includes English and Chinese versions, we extend it to the other four languages. **2)** For the creation of the new massive multilingual multimodal benchmark, we select and curate relevant data from the ScienceQA (Lu et al., 2022), MME (Fu et al., 2023), and SEED-Bench (Li et al., 2024a) datasets, adhering to established guidelines. These datasets are then transformed into a Visual Question Answering (VQA) format, resulting in a total of 12,000 samples across all six languages.

To mitigate potential errors and noise in the data acquisition process, we employ the following strategies to enhance the quality of our translations, as illustrated in Figure 3. First, we use GPT-4 to translate the original problem into the target language. Next, the initial translation is re-entered into for a re-check and refinement. This step helps identify and correct any immediate inconsistencies or inaccuracies. For manual calibration, we engage two groups of professional translators for each language involved in the study:

- **First Group for Refinement.** This group consists of three language experts who independently review and refine the translations generated by GPT-4. This results in three distinct versions of each translation.

- **Second Group for Voting.** The second group evaluates these refined translations and selects the most accurate one, ensuring it captures the intended meaning and nuances of the original text.

This calibration process significantly improves data quality by reducing errors and ensuring that translations are contextually accurate across languages. As a result, our benchmark achieves higher linguistic precision and cultural relevance, which we believe enhances the robustness of our research findings. Future versions will include additional details to further enhance readability and completeness.

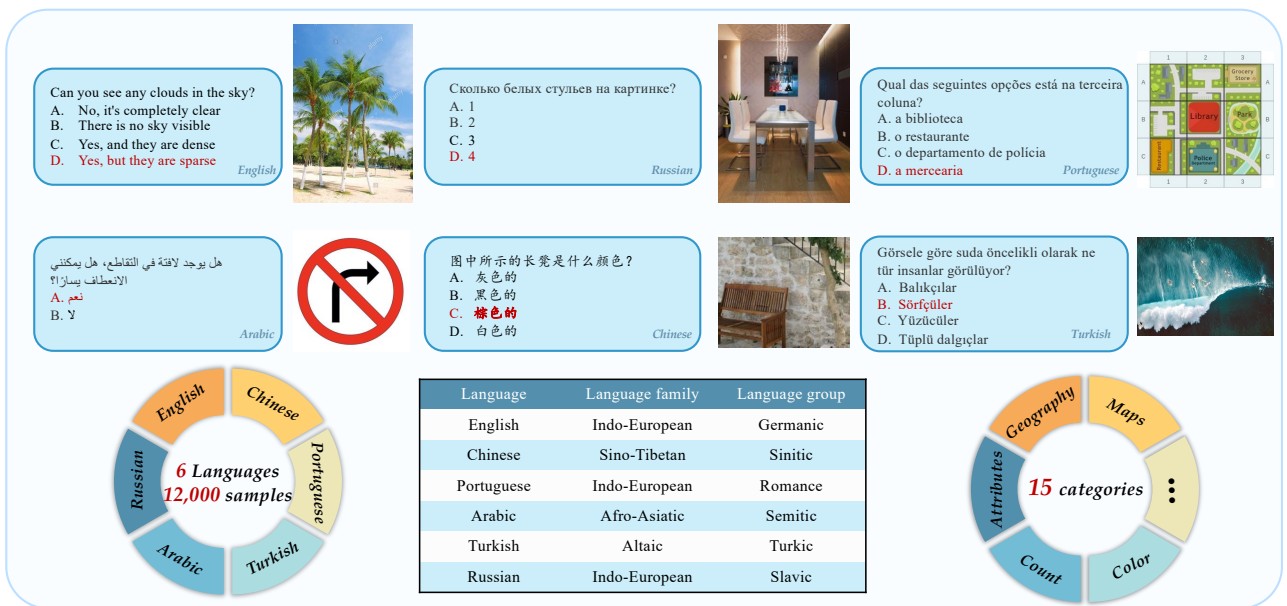

*Figure 4.* **The overview of MMMB benchmark.** It incorporates 6 languages, 15 categories, and 12,000 questions.

## 2.4. Evaluation Strategy

Since random guessing can lead to ∼25% Top-1 accuracy for 4-choice questions, it may reduce the discernible performance differences between various MLLMs. Additionally, MLLMs may have a tendency to favor a particular choice among the options (Liu et al., 2023c), further amplifying evaluation bias. To address these issues, we implement a circular validation strategy inspired by MMBench. Specifically, MMMB is adapted to the Yes/No question format, where each image is paired with two questions, requiring 'Yes' and 'No' answers, respectively. As shown in Figure 10 in Appendix, an answer is considered accurate only if both questions are answered correctly; failing either results in marking the entire instance as incorrect. This strategy ensures a more rigorous evaluation of MLLMs, reducing the impact of random guessing and promoting more robust comparisons across different models.

## 3. Methods

### 3.1. Preliminaries: Visual Instruction Tuning

A representative work in MLLMs is LLaVA (Liu et al., 2023b), which introduces a simple yet effective method for aligning the vision encoder and the pre-trained LLM. Specifically, for a given input image $\mathbf{X}_v$, LLaVA uses the pre-trained CLIP vision encoder ViT-L/14 (Radford et al., 2021) to extract visual features $\mathbf{Z}_v = g(\mathbf{X}_v)$. It then employs Vicuna (Chiang et al., 2023) as the LLM to generate textual embeddings $\mathbf{H}_t$. To align the vision encoder with the LLM, a projector, implemented as a multi-layer perceptron (MLP)

denoted by $\mathbf{W}$. This projector converts $\mathbf{Z}_v$ into language embedding tokens $\mathbf{H}_v$, effectively enabling the integration of multimodal information within the LLM's framework.

$$\mathbf{H}_v = \mathbf{W} \cdot \mathbf{Z}_v, \text{ with } \mathbf{Z}_v = g(\mathbf{X}_v). \quad (1)$$

Finally, we input $\mathbf{H}_v$ and $\mathbf{H}_t$ into LLM to generate the model's responses. However, after the modality alignment training, LLaVA loses its ability to process in non-English languages.

### 3.2. Pilot Study

To address the challenge of multilingual erosion in MLLMs due to the dominance of English in image-text data, we empirically observe an inherent mismatch between visual tokens $\mathbf{H}_v$ and textual tokens $\mathbf{H}_t$, which biases the model towards English semantics and increases the likelihood of English outputs. Specifically, the widely-used OpenAI-CLIP vision encoder (Radford et al., 2021), pre-trained on a large English-centric image-text corpus, yields visual representations more aligned with English.

To investigate this, we conduct the ablation study using OpenAI-CLIP and Chinese-CLIP (Yang et al., 2022). As shown in Figure 1, the OpenAI-CLIP model struggles with Chinese inputs, while the Chinese-CLIP model effectively processes and generates Chinese outputs. Additionally, to explore the distance between the visual features generated by different encoders and the Chinese prompts, we plot t-SNE visualizations for a more intuitive understanding. As illustrated in Figure 6, the t-SNE further reveal that the

visual features from Chinese-CLIP are more closely aligned with the Chinese prompts.

### 3.3. Textual Guidance to Drive Visual Token Alignment

First of all, we have to address a key challenge: due to the limited availability of non-English multimodal data, we cannot rely on extensive multilingual datasets to enhance the multilingual performance of MLLMs. Therefore, it is necessary to design a method that efficiently aligns visual and textual features at the language level, rather than relying on large multilingual data. Motivated by the pilot study, we aim to directly align visual tokens with textual embeddings at the language level. To this end, we propose PARROT, a novel method that leverages textual guidance to facilitate the multilingual alignment of visual features. PARROT enables the transition of English-biased visual features acquired through the OpenAI-CLIP to accommodate other languages, ensuring language-specific visual tokens are generated for LLMs based on multilingual inputs, thereby enhancing multilingual abilities.

First, we extract visual features through the vision encoder and transform them into language embedding tokens $\mathbf{H}_v$ using a projector. We obtain the embeddings $\mathbf{H}_t \in \mathbb{R}^{N \times C}$ derived from text inputs via the word embedding table. Subsequently, to convert the English-biased features into language-specific ones, we employ a cross-modal cross-attention mechanism to obtain $\mathbf{H}_v' \in \mathbb{R}^C$:

$$\mathbf{H}_v' = \text{Attention}(\mathbf{Q}, \mathbf{K}, \mathbf{V}) = \text{Softmax}\left(\frac{\mathbf{H}_v^{\text{cls}} \mathbf{H}_t^T}{\sqrt{C}}\right)\mathbf{H}_t, \tag{2}$$

where $\mathbf{Q} = \mathbf{H}_v$, and both $\mathbf{K}$ and $\mathbf{V}$ are equivalent to $\mathbf{H}_t$. $\mathbf{H}_v^{\text{cls}} \in \mathbb{R}^C$ represents the [CLS] token of $\mathbf{H}_v$. This process allows the visual features to be dynamically adjusted and transformed into language-specific semantic embeddings based on multilingual inputs.

Since the projected language embedding tokens $\mathbf{H}_v$ are biased towards English, we need to convert them into language-specific embeddings for different languages. To this end, we introduce a lightweight Mixture-of-Experts (MoE) module, which consists of a router and several language transformation experts. The MoE router is a linear layer that generates a probability distribution over the set of experts $\mathcal{E} = [e_1, e_2, \cdots, e_E]$, effectively selecting and activating specific experts. Each expert is an MLP designed to convert English-biased embeddings into language-specific embeddings. The inputs to experts $\mathcal{E}$ is $\mathbf{H}_v$, and the outputs have the same dimensions as the inputs.

Subsequently, to obtain a normalized probability distribution for activating language-specific experts, $\mathbf{H}_v'$ is fed as input to the router. The router network contains a linear layer that computes the normalized weight matrix using $\mathbf{H}_v'$ for

voting, producing $\mathcal{P} \in \mathbb{R}^E$:

$$\mathcal{P} = \text{Softmax}(\text{Linear}(\mathbf{H}_v')), \tag{3}$$

which selects and activates the appropriate experts. Next, we process the English-biased embeddings $\mathbf{H}_v$ through the selected experts to convert them into language-specific visual representations:

$$\text{MoE}(\mathbf{H}_v) = \sum_{i=1}^{k} \mathcal{P}[i] \cdot \mathcal{E}(\mathbf{H}_v)_i. \tag{4}$$

This approach effectively aligns English-biased embeddings with multiple languages, ensuring a more accurate and comprehensive representation across different linguistic contexts. To stabilize training and reduce the variance in visual-semantic information, ensuring the model performs well beyond the multilingual multimodal domain, we employ MoE reweighting to obtain the final language-specific visual embeddings $\mathbf{G}_v$:

$$\mathbf{G}_v = \mathbf{H}_v + \alpha\text{MoE}(\mathbf{x}), \tag{5}$$

where $\alpha$ is a trade-off parameter. In conclusion, we first fuse the visual and textual inputs via Eq. 2 to transform the visual embeddings with textual guidance. The fused result is then input into the MoE module, which selects and activates the most relevant language experts via Eq. 3 to obtain language-specific embeddings as shown in Eq. 4. Finally, MoE reweighting is applied to convert visual embeddings with less variance in original visual-semantic information 5. This approach enables the MLLM to gain multilingual capabilities using minimal multilingual data. Figure 5 illustrates the architecture, the detailed MoE module, and the training stages of PARROT.

### 3.4. Training Stage

Our goal is to enhance the multilingual capabilities of MLLMs with minimal multilingual data. The training procedure is divided into two distinct stages:

**Stage 1: Modality Alignment.** In this stage, we freeze both the vision encoder and the LLM weights, focusing solely on optimizing the projectors to bridge the modality gap. Notably, the MoE module is bypassed entirely, meaning the image tokens do not pass through the MoE since the primary goal of this stage is to train the projector using a large number of image-text pairs. This enables the projector to align image tokens and textual tokens effectively without interference from the untrained MoE module.

**Stage 2: Instruction Tuning for Multilingual Alignment.** We continue to freeze the vision encoder weights while training all other modules. In this stage, we introduce multilingual training data and randomly initialize the parameters

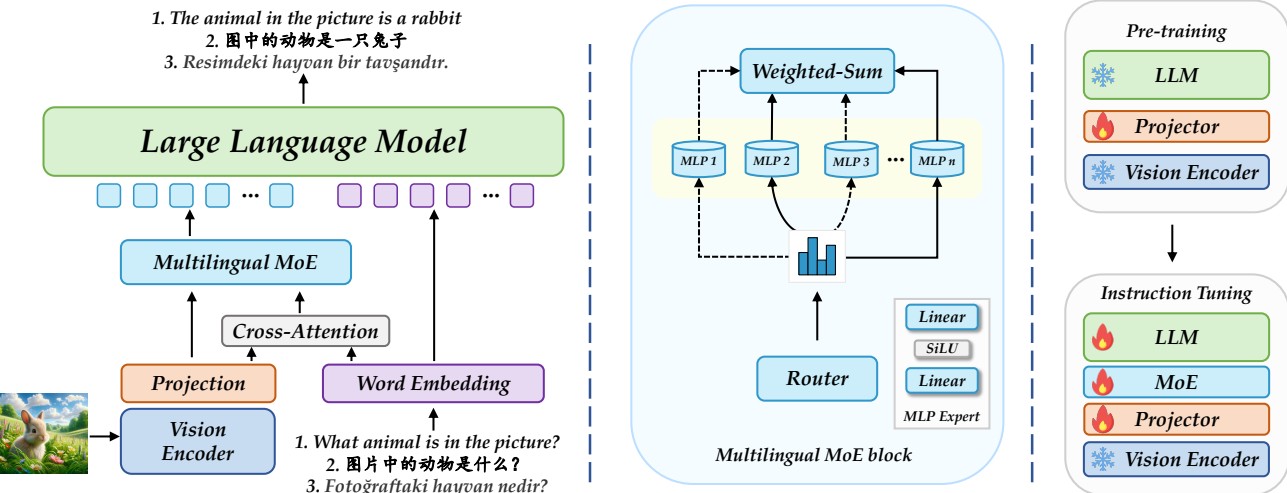

Figure 5. **The overall architecture of** PARROT. It converts English-biased features to language-specific features based on the multilingual MoE module, aiming to improve the multilingual capabilities. The training details within each stage are presented on the right.

of MoE. The MoE is optimized with textual guidance, which drives the alignment of visual tokens while leveraging the well-trained projector. The prior alignment achieved in the pre-training stage facilitates efficient optimization of the MoE during this phase. The entire training process of PAR-ROT is outlined in pseudocode, as shown in Algorithm 1 in the Appendix.

To address data scarcity in non-English languages, we use a semi-automatic method similar to the one depicted in Figure 3 to acquire image-text data. We randomly partition the ShareGPT4V dataset (Chen et al., 2023b) for each language, extracting non-duplicate, non-parallel image-text pairs for training, ultimately obtaining nearly 10K samples per language. This two-stage training approach ensures effective modality and multilingual alignment, even with limited non-English data, aligning well with the challenges of data scarcity in low-resource languages.

## 4. Experiments

In this section, we begin with an overview of the experimental framework, providing details on specific implementations, evaluation benchmarks, and MLLMs used for comparative evaluation. Following this, we conduct a comprehensive comparison of PARROT with the state-of-the-art approaches using multilingual benchmarks. We also compare PARROT with leading models across a range of multimodal tasks. Finally, we conclude with ablation studies and visualization of multilingual cases, highlighting the exceptional ability of PARROT in handling multilingual tasks.

### 4.1. Experimental Setup

**Implementation Details.** In this study, we configure PARROT with the pre-trained CLIP ViT-L/14 (Radford et al.,

2021) as the vision encoder. To validate the effectiveness of PARROT, we select both Qwen1.5 and Qwen2 (Bai et al., 2023a) as the backbones. The initial learning rates for the two stages are set at $1e^{-3}$ and $2e^{-5}$, respectively, with the batch size of 256 and 128. The entire training process is optimized to 21 hours on the 16×A100 GPUs setup, benefiting from the relatively small training datasets. Additionally, BF16 and TF32 precision formats are employed to balance speed and accuracy throughout the training process. As defined in Eq. 4, we set the number of experts to six to correspond with the number of languages. Each expert is an MLP composed of two linear layers with SiLU (Elfwing et al., 2018) activation function. More details are provided in Table 13.

**Evaluation Benchmark.** Our evaluation consists of two parts: one assessing the multilingual capabilities of MLLMs, while the other evaluating its overall performance. The first part is conducted on two datasets: multilingual MM-Bench (Liu et al., 2023c) and a newly developed benchmark MMMB. The second part of the evaluation spans a wide range of multimodal tasks, such as MME (Fu et al., 2023), MMStar (Chen et al., 2024), ScienceQA (Lu et al., 2022), RealWorldQA (x.ai, 2024) and SEED-Bench (Li et al., 2024a), with performance visualized in a radar chart in Figure 7b.

**Comparison Models.** For comprehensive comparisons, we select leading open-source models in MLLMs, including LLaVA-1.5 (Li et al., 2023a), LLaVA-NeXT (Liu et al., 2024), Monkey (Li et al., 2023d), VisualGLM (Du et al., 2022), VisCPM (Hu et al., 2023), GLM-4v (GLM et al., 2024), ShareGPT4V (Chen et al., 2023b), InstructBLIP (Dai et al., 2023), mPLUG-Owl2 (Ye et al., 2023), Idefics3 (Laurençon et al., 2024), LLaVA-OneVision (Li et al., 2024b), and Qwen2-VL (Wang et al., 2024a). For the evaluation

*Table 1.* Accuracy performance comparison on multilingual benchmarks. We report all compared methods with VLMEvalKit (Duan et al., 2024). The best and second results are shown in **bold** and underline, respectively.

| Method | LLM | MMMB | | | | | | MMBench | | | | | |
|---|---|---|---|---|---|---|---|---|---|---|---|---|---|
| | | *en* | *zh* | *pt* | *ar* | *tr* | *ru* | *en* | *zh* | *pt* | *ar* | *tr* | *ru* |
| VisualGLM (Du et al., 2022) | ChatGLM-6B | 31.05 | 18.07 | 19.42 | 15.38 | 22.81 | 19.77 | 23.2 | 17.18 | 11.43 | 2.92 | 6.62 | 5.33 |
| VisCPM-Chat (Hu et al., 2023) | CPM-Bee-10B | 53.10 | 47.54 | 28.19 | 26.90 | 26.78 | 26.84 | 45.88 | 46.39 | 15.81 | 1.46 | 9.19 | 1.20 |
| Qwen-VL-Chat (Bai et al., 2023b) | Qwen-7B | 56.02 | 57.77 | 46.37 | 43.04 | 41.05 | 48.65 | 54.29 | 56.52 | 43.12 | 35.73 | 39.17 | 42.86 |
| InstructBLIP (Dai et al., 2023) | Vicuna-7B | 39.47 | 32.92 | 35.67 | 23.80 | 28.36 | 36.37 | 27.83 | 18.81 | 27.14 | 3.26 | 8.50 | 20.87 |
| mPLUG-Owl2 (Ye et al., 2023) | LLaMA2-7B | 67.25 | 60.99 | 59.70 | 45.78 | 45.43 | 62.63 | 66.15 | 59.36 | 58.24 | 37.88 | 47.68 | 60.39 |
| Monkey (Li et al., 2023d) | Qwen-VL-7B | 66.02 | 58.18 | 46.31 | 38.83 | 37.66 | 48.59 | 58.07 | 53.52 | 49.57 | 31.01 | 31.35 | 45.18 |
| LLaVA-1.5 (Liu et al., 2023a) | Vicuna-v1.5-7B | 67.07 | 58.83 | 59.76 | 43.50 | 46.43 | 59.06 | 65.37 | 58.33 | 59.02 | 36.16 | 43.90 | 56.95 |
| LLaVA-NeXT (Liu et al., 2024) | LLaMA3-8B | 70.92 | 64.33 | 63.21 | 48.34 | 48.02 | 66.35 | 69.80 | 63.31 | 61.83 | 47.64 | 47.03 | 64.99 |
| DeepSeek-VL-7B (Lu et al., 2024a) | Deepseek-7B | 72.66 | 65.95 | 64.41 | 49.70 | 49.07 | 67.57 | 70.71 | 64.03 | 62.61 | 48.05 | 47.95 | 65.53 |
| GLM-4v-9B (GLM et al., 2024) | GLM-4-9B | 69.26 | 62.83 | 61.59 | 47.23 | 46.9 | 64.35 | 67.91 | 61.31 | 60.01 | 46.13 | 45.7 | 63.09 |
| ShareGPT4V (Chen et al., 2023b) | Vicuna-v1.5-7B | 69.24 | 60.23 | 60.29 | 43.57 | 45.26 | 61.23 | 69.59 | 61.6 | 59.62 | 37.37 | 43.38 | 59.45 |
| Idefics3-8B (Laurençon et al., 2024) | LLaMA3-8B | 74.50 | 67.70 | 66.08 | 50.91 | 50.41 | 69.66 | 73.50 | 66.79 | 65.53 | 49.85 | 49.79 | 68.68 |
| Qwen2-VL (Wang et al., 2024a) | Qwen2-7B | **80.51** | **80.23** | 78.11 | 74.07 | 71.72 | 79.33 | **78.59** | **78.39** | 75.93 | 74.70 | 73.45 | 75.88 |
| LLaVA-OneVision (Li et al., 2024b) | Qwen2-7B | 79.03 | 78.23 | 75.91 | 73.36 | 67.79 | 76.37 | 76.74 | 75.30 | 73.45 | 70.44 | 64.85 | 73.14 |
| PARROT | Qwen1.5-7B | 70.00 | 68.13 | 67.31 | 62.69 | 58.01 | 66.26 | 70.70 | 70.36 | 65.12 | 57.82 | 58.43 | 64.00 |
| PARROT | Qwen2-7B | 80.11 | 80.03 | **79.62** | **76.55** | **75.02** | **79.94** | 78.02 | 77.16 | **76.76** | **75.92** | **74.05** | **77.71** |

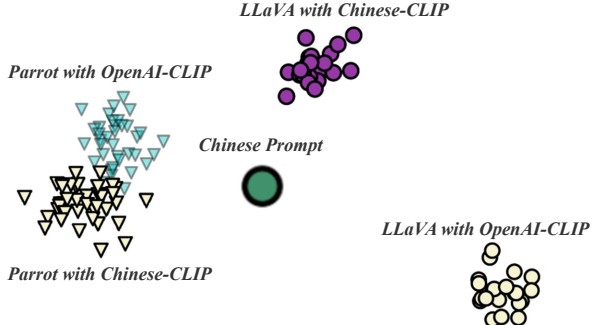

*Figure 6.* t-SNE visualizations of LLaVA and PARROT using different vision encoders.

process, we employ the VLMEvalKit (Duan et al., 2024) in OpenCompass, ensuring consistent configuration settings across all methods to maintain fairness in comparison.

### 4.2. Main Results

As shown in Table 1, PARROT-Qwen2-7B achieves state-of-the-art (SOTA) performance across four languages in both the MMBench and MMMB benchmark, with English and Chinese in second place. Figure 7b demonstrates that PARROT excels not only in multilingual capabilities but also in handling complex multimodal tasks, such as MME (Fu et al., 2023), MMStar (Chen et al., 2024), and SEED-Bench (Li et al., 2024a). In Figure 7c, we infer PARROT using a Chinese prompt and visualize the expert distributions within the MoE, revealing the dynamic activation of different experts for different languages. Additionally, as illustrated in Figure 4, PARROT achieves competitive performance in existing multilingual benchmarks, utilizing **less than 1%** of the data compared to other multilingual MLLMs.

To validate the effectiveness of the PARROT architecture, we use a Chinese prompt (translated to English as "please describe the image in detail"), sample 50 images of 'tiger

cat' from ImageNet, and perform t-SNE (Van der Maaten & Hinton, 2008) visualizations of both visual and textual features in LLaVA and PARROT. As shown in Figure 6, for LLaVA, the Chinese-CLIP-based visual features are significantly closer in high-dimensional space compared to OpenAI-CLIP. However, thanks to the architecture we designed, PARROT bridges the gap with textual features effectively, regardless of the vision encoder used.

### 4.3. Further Analysis

In this section, we explore several critical questions to comprehensively analyze PARROT.

- **Are all components of PARROT equally effective?** As shown in Figure 7a, incorporating multilingual data enhances performance across all languages. Additionally, the MoE module contributes significantly to performance improvements, validating the effectiveness of our proposed method.

- **How does PARROT handle the quality and imbalance challenges?** Large-scale translated multilingual data, while seemingly abundant, often suffers from quality issues (*e.g.*, translation errors, cultural mismatches, noisy artifacts), particularly for low-resource languages. Even with massive data volumes, low-resource languages often remain underrepresented as high-resource languages dominate training distributions (*e.g.*, 90%+ of tokens in typical datasets). Forcing higher proportions of low-resource data can risk triggering the "curse of multilingualism," where models sacrifice high-resource language performance to accommodate low-resource languages, as observed in prior work (Conneau et al., 2019; Chang et al., 2023; Blevins et al., 2024). Our approach strategically prioritizes high-quality alignment signals over raw data quantity. This avoids the pitfalls of noisy translation and

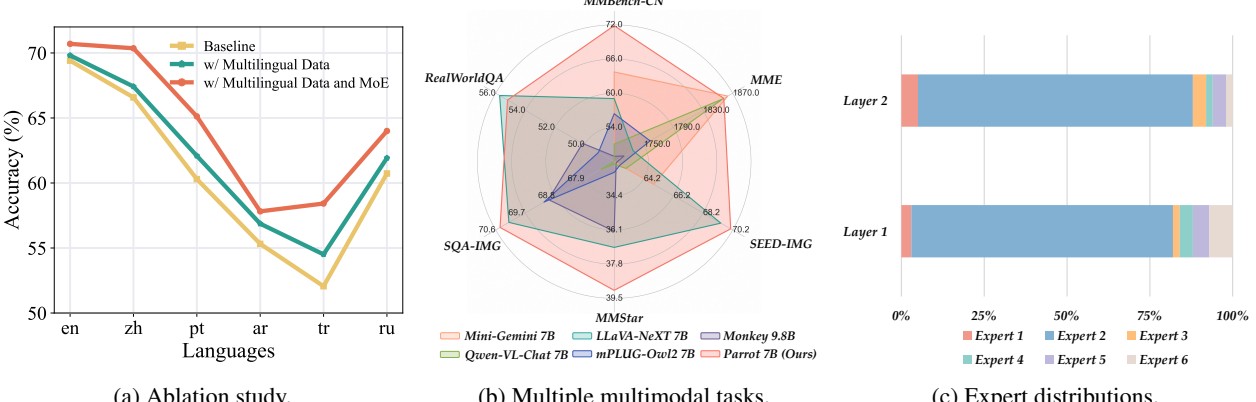

(a) Ablation study.     (b) Multiple multimodal tasks.     (c) Expert distributions.

*Figure 7.* **Left:** The ablation study of multilingual data and the MoE module using the MMBench benchmark. **Middle:** The performance of PARROT on a broad range of multimodal tasks compared with existing models. **Right:** Expert distributions of MoE. We summarize the activated experts during the feed-forward process using Chinese Prompts.

the imbalance inherent to brute-force multilingual scaling, ensuring stable performance across all languages without compromising high-resource capabilities.

- **How does PARROT perform compared to a naive, translation-based baseline?** To assess this, we implement a baseline using the Google Translation API that translates non-English queries to English, processes them, and translates responses back to the original language. As shown in Table 10, the results reveal a "seesaw effect": while this naive approach yields improvements in certain languages like Chinese, it simultaneously causes performance degradation in others, particularly Russian and Portuguese. This phenomenon highlights the fundamental limitations of relying solely on translation services for addressing multilingualism in multimodal tasks, underscoring the need for more sophisticated approaches like PARROT.

- **Are the scaling laws effective in PARROT?** To explore the effectiveness of scaling laws in multilingual settings, we conduct experiments where the multilingual data (excluding Chinese and English) is gradually expanded until it matches the volume of Chinese data (70K). As shown in Table 11, the results indicate that PARROT continues to follow the multilingual scaling law. For example, the performance on Portuguese increased by 3.0 points, and Arabic saw a 5.2-point gain. Additionally, PARROT benefits from model size scaling, as shown in Table 12.

- **Do the main performance gains come from the multilingual dataset?** As shown in Table 9 and Figure 7a, LLaVA shows limited improvements with the addition of multilingual data. In contrast, PARROT achieves substantial gains, significantly outperforming LLaVA. Therefore, it is ensure that the primary performance gains come from the design of PARROT.

### 4.4. Visualization of Multilingual Conversations

To enhance the intuitive understanding of the PARROT's multilingual capability, we prepare a comprehensive case study accompanied by illustrative visuals. For instance, as depicted in Figure 8, our framework demonstrates remarkable multilingual capabilities. This underscores the PARROT's versatility in navigating different languages and presents its potential in bridging linguistic gaps across diverse domains. Through careful analysis and visualization, we aim to provide a deeper insight into the mechanism driving this capability, illustrating its practical implications and potential applications in real-world scenarios. This visualization serves as a strong indicator of the PARROT's solid architecture and its exceptional ability to understand, process, and generate multiple languages with remarkable efficiency. More multilingual conversation cases are shown in Appendix F.

## 5. Conclusion

This paper addresses the critical challenge of improving the multilingual capabilities of MLLMs and investigates the misalignment of visual features across languages. We introduce PARROT, a novel approach that leverages textual guidance to align visual tokens at the language level, enabling the conversion of English-biased visual embeddings into language-specific ones through an MoE module. Extensive experiments conducted on a newly introduced benchmark, the Massive Multilingual Multimodal Benchmark (MMMB), across six languages demonstrate that PARROT achieves state-of-the-art performance compared to existing methods, with particularly notable improvements in Turkish and Arabic. PARROT not only advances the frontier of MLLMs but also highlights the importance of equitable access to technological benefits across linguistic and cultural contexts.

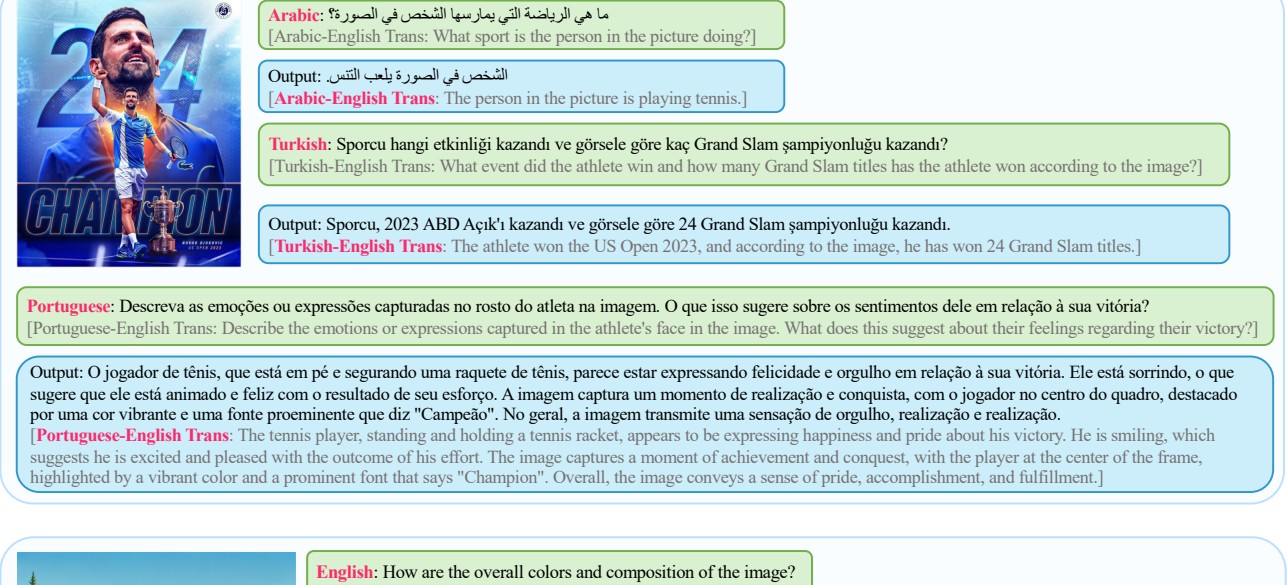

*Figure 8.* Multimodal conversation cases of PARROT in multiple languages.

## Acknowledgments

This work is partially supported by National Key R&D Program of China (2024YFE0202800), NSFC (62376118, 62476123), Key Program of Jiangsu Science Foundation (BK20243012), Fundamental Research Funds for the Central Universities (2024300373, 14380021), the AI & AI for Science Project of Nanjing University, Collaborative Innovation Center of Novel Software Technology and Industrialization.

## Impact Statement

PARROT leveraging MoE to enhance multilingual alignment presents a positive social impact by promoting linguistic diversity and inclusivity. To address the challenge of the imbalanced language data in SFT datasets and improve non-English visual tokens alignment, this approach contributes to breaking language barriers and facilitating cross-cultural communication, thereby fostering understanding and collaboration across diverse linguistic communities. Additionally, the creation of the Massive Multilingual Multimodal Benchmark (MMMB) fills a crucial gap in evaluating multilingual capabilities, enabling researchers to assess and improve upon models' performance across different languages and cultures. This paper presents work whose goal is to advance the field of Machine Learning and the potential societal impact of this work is largely positive.

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

# A. The Details of Training Datasets

In this section, we analyze the multilingual data in LLaVA (Liu et al., 2023b). From Table 2 and Figure 9, it is evident that during the pre-train stage, LLaVA solely utilizes multimodal image-text pairs data for training, comprising 558K of English data. During the SFT stage, both multimodal and text-only data are incorporated into the training process. Multilingual data appear only in the text-only dataset. Apart from English, the most prominent non-English data is Chinese, amounting to just 3.1K, constituting 0.25% of the total dataset. Therefore, it is evident that LLaVA's datasets are English-centric and imbalanced. The specific language and abbreviation are as follows: English (*en*), Chinese (*zh*), Korean (*ko*), Spanish (*es*), French (*fr*), Japanese (*ja*), German (*de*), Portuguese (*pt*), Traditional Chinese (*zh-tw*), Italian (*it*).

*Table 2.* The detailed information about LLaVA's datasets.

(a) The language information in two stages.

| Training Stage | Type | Total Size | English | Other Languages |
|---|---|---|---|---|
| Stage 1 (Pre-train) | Multimodal | 558K | 558K | - |
| | Text-only | - | - | - |
| Stage 2 (SFT) | Multimodal | 624K | 558K | - |
| | Text-only | 41K | 31K | 10K |

(b) The top-10 multilingual information

| Language | *en* | *zh* | *ko* | *es* | *fr* | *ja* | *de* | *pt* | *zh-tw* | *it* |
|---|---|---|---|---|---|---|---|---|---|---|
| Size | 31K | 3192 | 1219 | 1123 | 1049 | 551 | 435 | 422 | 305 | 234 |

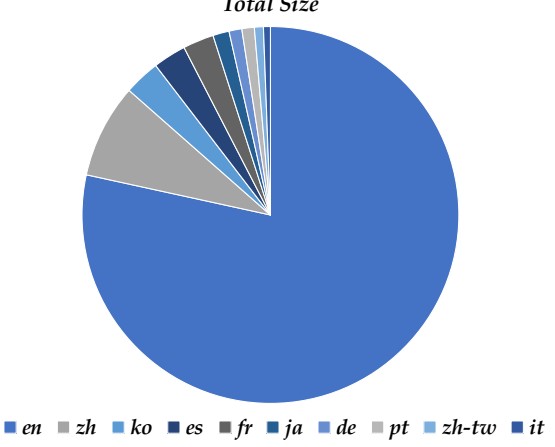

*Total Size*

■ *en* ■ *zh* ■ *ko* ■ *es* ■ *fr* ■ *ja* ■ *de* ■ *pt* ■ *zh-tw* ■ *it*

*Figure 9.* The pie chart of LLaVA's multilingual data.

*Figure 10.* An example of circular evaluation strategy.

# B. Related Work

**Multimodal Large Language Models.** The domain of MLLMs has witnessed significant advances, particularly in the enhancement of visual and language processing. Current MLLMs are usually a combination of visual encoders (Radford et al., 2021; Sun et al., 2023; Fang et al., 2023; Zhang et al., 2022; Oquab et al., 2023; Zhai et al., 2023), LLMs, and fusion modules. Innovations like Flamingo (Alayrac et al., 2022) have advanced visual representation by integrating a Perceiver Resampler with vision encoders. BLIP-2 (Li et al., 2023b) and InstructBLIP (Dai et al., 2023) employ Q-Former to connect the frozen LLM and vision encoder. InternVL (Chen et al., 2023c) trains huge ViT and QFormer to integrate visual modalities through a multi-stage training method. MiniGPT4 (Zhu et al., 2023) leverages both a Q-Former and a linear projector to bridge the gap between the vision module and LLM. Furthermore, LLaVA (Liu et al., 2023b) adopts

*Table 3.* Details on the PARROT's training data, which is derived from publicly available datasets.

| Training Stage | Datasets | Samples | Total |
|---|---|---|---|
| Stage 1 | LLaVA-1.5-pretrain (Liu et al., 2023b) | 558K | |
| | Laion-Caption (Schuhmann et al., 2022) | 12K | 1.2M |
| | CC12M-Caption (Changpinyo et al., 2021) | 645K | |
| Stage 2 | LLaVA-1.5-finetune (Liu et al., 2023b) | 665K | |
| | ShareGPT4V-zh (Chen et al., 2023b) | 71K | |
| | ShareGPT4V-pt (Chen et al., 2023b) | 14K | |
| | ShareGPT4V-ar (Chen et al., 2023b) | 12K | 793K |
| | ShareGPT4V-tr (Chen et al., 2023b) | 17K | |
| | ShareGPT4V-ru (Chen et al., 2023b) | 14K | |

a simple MLP projector to promote the alignment between the LLM and vision encoder. mPLUG-Owl (Ye et al., 2023) introduces an approach that begins to finetune the vision encoder and align visual features, followed by tuning the LLM using LoRA (Hu et al., 2021). Qwen-VL (Bai et al., 2023b) improves visual module resolution to 448, aiming to refine the model's visual processing capabilities. Fuyu-8B (Bavishi et al., 2023) directly projects image patches before integration with LLM. MM1 (McKinzie et al., 2024) has conducted ablative studies on connector design choices, revealing that the modality adapter type is less critical than the number of visual tokens and the resolution. MiniGemini (Li et al., 2024d) utilizes high-resolution visual tokens and high-quality data to narrow the performance gap with GPT-4 and Gemini. With the rapid advancements in open-source models, proprietary models such as GPT-4o (OpenAI, 2024), Gemini (Team et al., 2023; Reid et al., 2024), Qwen-VL-series (Wang et al., 2024a), and Claude3 (anthropic, 2024) have achieved outstanding results in evaluations and practical applications. Some other recent works (Lu et al., 2024b; Zhu et al., 2025; Sun et al., 2025c; Li et al., 2025; Sun et al., 2025a; Dong et al., 2025; Zhang et al., 2024c) provide valuable insights and future directions for building and understanding vision-language models. In this work, owing to the simplicity of the LLaVA architecture, we adopt a framework similar to LLaVA to design our model.

**Multilingual Multimodal Models.** Recent years have witnessed rapid progress in the expansion of multimodal models to include a wider variety of languages. $M^3P$ (Ni et al., 2021) leverages English as a pivot and alternates between English-only vision-language pre-training and multilingual masked language modeling. In contrast, $UC^2$ (Zhou et al., 2021) translates English captions into various languages and uses images as the anchor. mCLIP (Chen et al., 2023a) enhances the CLIP model by aligning it with a multilingual text encoder through knowledge distillation. Thanks to the expansion of the overall capabilities of large language models (AI, 2024; Bai et al., 2023a; Jiang et al., 2023; Young et al., 2024), their multilingual capacities have significantly improved. Integrating multilingual LLMs with visual abilities has increasingly become a research focus. In the domain of LLMs, PaLI (Chen et al., 2022) develops a 17B multilingual language-image model that spans over 100 languages. Ying-VLM (Li et al., 2023c) discovers that instruction tuning in English can extend its applicability to other languages. Ziya-Visual (Lu et al., 2023) illustrates the translation of English image-text datasets into Chinese, using in-context learning for instruction-response generation. VisCPM (Hu et al., 2023) introduces a training paradigm that fine-tunes the MLLM in a quasi-zero-shot manner based on a strong multilingual large language model. Despite these advancements, they are primarily confined to two languages or rely on the massive translated corpus. On the other hand, there is no suitable multilingual benchmark for MLLMs to evaluate the performance of multiple languages. There are also some multilingual research studies in other domains, such as multilingual machine translation (Zhao et al., 2024; Pires et al., 2023; Purason & Tättar, 2022; Zhang et al., 2021).

## C. Additional Experimental Results

In this section, we present additional experiments and ablation studies to further validate the generality and capability of PARROT across various tasks. Additionally, we elaborate on the training details for Figure 1 to provide a clearer understanding.

### C.1. Bilingual Evaluation on LLaVA-Bench

VisCPM (Hu et al., 2023) extends the LLaVA-Bench dataset to the Chinese version for bilingual evaluation. To comprehensively compare PARROT with other multilingual models, we conduct experiments on this benchmark. Due to the deprecation

*Table 4.* Experimental results on LLaVA Test Set accessed by GPT-4. Con: Conversation, DD: Detailed Description, CR: Complex Reasoning, AVG: the average score of three tasks. The symbol denotes that the data are judged following the version of GPT-4-1106-preview because the GPT-4-0314 version is deprecated by OpenAI.

| Model | | LLM Backbone | English | | | | Chinese | | | |
|---|---|---|---|---|---|---|---|---|---|---|
| | | | Con | DD | CR | AVG | Con | DD | CR | AVG |
| English Model | MiniGPT-4 | Vicuna-13B | 65.0 | 67.3 | 76.6 | 69.7 | - | - | - | - |
| | InstructBLIP | Vicuna-13B | 81.9 | 68.0 | 91.2 | 80.5 | - | - | - | - |
| | LLaVA | Vicuna-13B | 89.5 | 70.4 | 96.2 | 85.6 | - | - | - | - |
| En-Zh Bilingual Model | mPLUG-OWL | BLOOMZ-7B | 64.6 | 47.7 | 80.1 | 64.2 | 76.3 | 61.2 | 77.8 | 72.0 |
| | VisualGLM | ChatGLM-6B | 62.4 | 63.0 | 80.6 | 68.7 | 76.6 | 87.8 | 83.6 | 82.7 |
| | Qwen-VL-Chat | Qwen-7B | 82.4 | 76.9 | 91.9 | 83.8 | 82.3 | 93.4 | 89.5 | 88.2 |
| | VisCPM-Balance | CPM-Bee-10B | 75.5 | 64.7 | 91.3 | 77.3 | 85.4 | 81.4 | 96.6 | 88.0 |
| Multilingual Model | PARROT | Qwen1.5-7B | 82.5 | 71.0 | 89.3 | 81.1 | 82.1 | 88.6 | 92.3 | 87.7 |

of the GPT-4-0314 version by OpenAI, we test PARROT in LLaVA-Bench following the version of GPT-4-1106-preview for comparison. As shown in Table 4, PARROT not only demonstrates exceptional ability in the English version of this benchmark but also presents competitive performance in the Chinese version.

Notably, as shown in Table 5, VisCPM uses 140M English data and 1M Chinese data to train the model. In comparison, Qwen-VL-Chat uses 1.1B English data and 300M Chinese data, whereas PARROT only utilizes approximately 2M data in total. Despite using less than 1% of the training data, PARROT achieves remarkable performance in both the English and Chinese versions on LLaVA-Bench. Owing to the architecture we proposed, significant improvement in the model's multilingual capability can be achieved with minimal data usage.

*Table 5.* Comparison of vision encoders, LLMs, and training data in different models.

| Model | vision encoder | LLM | Training Data |
|---|---|---|---|
| mPLUG-Owl | ViT-L/14 (0.3B) | BLOOMZ-7B | - |
| VisualGLM | Q-Former (1.6B) | ChatGLM-6B | English: 300M; Chinese 30M |
| Qwen-VL-Chat | ViT-bigG (1.9B) | Qwen-7B | English: 1.1B; Chinese: 300M |
| VisCPM | Muffin (0.7B) | CPM-Bee-10B | English: 140M; Chinese: 1M |
| PARROT | ViT-L/14 (0.3B) | Qwen1.5-Chat-7B | English: 1.8M; Chinese: 71K |

## C.2. Further Ablation Studies

**Ablation study on each component.** We conduct an ablation experiment on the multilingual data and the MoE module. As shown in Figure 7a, using multilingual data improves performance in each language. Moreover, the MoE module significantly improves performance, demonstrating the effectiveness of our proposed method.

**Ablation study on different datasets.** As shown in Table 6, it is evident that the inclusion of different multilingual datasets continually improves performance on the MMBench benchmark, and all models with 7B parameters are used for this experiment. This highlights the robustness and scalability of our approach to handling multiple languages effectively.

**Ablation study on monolingual fine-tuning datasets.** The ablation study presented in Table 14 evaluates the performance of different monolingual datasets added incrementally to the baseline dataset LLaVA-1.5-finetune. It highlights the significant impact of adding different multilingual datasets to a baseline model. Each dataset incrementally improves performance in its respective language and, when combined, leads to overall enhanced performance across all evaluated languages. This indicates the robustness and effectiveness of the proposed method in handling multilingual data, making it a scalable solution for multilingual tasks.

*Table 6.* Ablation study on different multilingual training datasets in MMBench benchmark. Models with 7B parameters are used for this ablation.

| Dataset | English | | Chinese | | Portuguese | | Arabic | | Turkish | | Russian | |
|---|---|---|---|---|---|---|---|---|---|---|---|---|
| LLaVA-1.5-finetune | 69.4 | | 66.6 | | 60.3 | | 55.3 | | 52.1 | | 60.7 | |
| + zh | 69.2 | -0.2 | 68.6 | +2.0 | 64.1 | +3.8 | 59.1 | +3.8 | 50.9 | -1.2 | 61.6 | +0.9 |
| + zh pt | 71.1 | +1.7 | 70.4 | +3.8 | 65.4 | +5.1 | 57.9 | +2.6 | 52.1 | +0.0 | 62.9 | +2.2 |
| + zh pt ar | 71.0 | +1.6 | 68.6 | +2.0 | 65.7 | +5.4 | 58.6 | +3.3 | 52.2 | +0.1 | 62.2 | +1.5 |
| + zh pt ar tr | 70.4 | +1.0 | 68.7 | +2.1 | 64.9 | +4.6 | 61.2 | +5.9 | 59.7 | +7.6 | 62.0 | +1.3 |
| + zh pt ar tr ru | 70.7 | +1.3 | 70.4 | +3.8 | 65.1 | +4.8 | 57.8 | +2.5 | 58.4 | +6.3 | 64.0 | +3.3 |

## C.3. Comparison of Different Vision Encoders

We also compare the different vision encoders within the PARROT framework in Table 7. It shows that the Chinese-CLIP-based model maintains comparable multilingual performance to the OpenAI-CLIP-based one. This demonstrates that our framework can be compatible with different vision encoders and achieve multilingual alignment through the MoE module.

*Table 7.* The comparison of various vision encoders within the PARROT framework.

| Method | LLM | Vision Encoder | MMMB | | | | | | MMBench | | | | | |
|---|---|---|---|---|---|---|---|---|---|---|---|---|---|---|
| | | | en | zh | pt | ar | tr | ru | en | zh | pt | ar | tr | ru |
| LLaVA-1.5 | Vicuna-v1.5-7B | OpenAI-CLIP | 67.07 | 58.83 | 59.76 | 43.50 | 46.43 | 59.06 | 65.37 | 58.33 | 59.02 | 36.16 | 43.90 | 56.95 |
| LLaVA-1.5 | Vicuna-v1.5-7B | Chinese-CLIP | 66.45 | 59.23 | 59.22 | 42.68 | 46.11 | 58.89 | 65.92 | 57.85 | 58.45 | 36.90 | 44.82 | 56.32 |
| ShareGPT4V | Vicuna-v1.5-7B | OpenAI-CLIP | 69.24 | 60.23 | 60.29 | 43.57 | 45.26 | 61.23 | 69.59 | 61.60 | 59.62 | 37.37 | 43.38 | 59.45 |
| ShareGPT4V | Vicuna-v1.5-7B | Chinese-CLIP | 68.65 | 60.85 | 59.49 | 44.33 | 44.90 | 61.88 | 70.28 | 61.91 | 58.83 | 37.00 | 42.55 | 58.97 |
| PARROT | Qwen1.5-7B | OpenAI-CLIP | 70.00 | 68.13 | 67.31 | 62.69 | 58.01 | 66.26 | 70.70 | 70.36 | 65.12 | 57.82 | 58.43 | 64.00 |
| PARROT | Qwen1.5-7B | Chinese-CLIP | 69.22 | 69.24 | 66.32 | 62.15 | 57.77 | 64.31 | 69.95 | 70.87 | 64.92 | 56.57 | 57.13 | 63.15 |

*Table 8.* The performance of different vision encoders and LLMs on MMBench and MMMB. MMB refers to MMBench. "En/en" represents the English version, and "CN/zh" represents the Chinese version.

| Method | Vision encoder | LLM | MMB-EN | MMB-CN | MMMB-en | MMMB-zh |
|---|---|---|---|---|---|---|
| LLaVA | OpenAI-CLIP ViT-L/14 | Vicuna 7B | 65.4 | 58.3 | 67.1 | 58.8 |
| LLaVA | OpenAI-CLIP ViT-L/14 | Qwen1.5-Chat 7B | 68.8 | 66.4 | 68.2 | 62.4 |
| LLaVA | Chinese-CLIP ViT-L/14 | Qwen1.5-Chat 7B | 68.1 | 68.3 | 67.6 | 66.1 |
| PARROT | OpenAI-CLIP ViT-L/14 | Qwen1.5-Chat 7B | 70.7 | 70.4 | 70.0 | 68.1 |

## C.4. Comparison with LLaVA using the Same Data

To validate the effectiveness of our proposed approach, we conduct further experiments with an ablation study. Specifically, we expand the baseline LLaVA method by incorporating the same multilingual data used in PARROT. Both models are evaluated on the MMMB dataset, and the results are presented in the Table 9. From the results, we observe that while LLaVA shows a slight improvement with the addition of multilingual data, the increase in performance is limited. In contrast, our PARROT model demonstrates a substantial improvement when multilingual data is included, significantly outperforming LLaVA. This highlights that simply adding multilingual data is not sufficient to bridge the multilingual gap, further emphasizing the effectiveness of our proposed design.

## C.5. Data Scaling and Model Size Scaling

To further investigate the scaling law in multilingual settings, we have conducted experiments where we progressively expanded the multilingual data (excluding Chinese and English) until it reached a volume comparable to the amount of Chinese data (~70K). The results, shown in the Table 11, demonstrate that PARROT still satisfies the multilingual scaling law. For instance, the performance on Portuguese improved by 3.0 points, and Arabic saw a gain of 5.2 points. As we

*Table 9.* The comparison of the baseline LLaVA and PARROT using the same multilingual training data. LLaVA shows limited improvement with multilingual data, while PARROT achieves significant gains, greatly outperforming LLaVA.

| Method | MMMB | | | | | |
|---|---|---|---|---|---|---|
| | *en* | *zh* | *pt* | *ar* | *tr* | *ru* |
| LLaVA w/o Multilingual data | 67.1 | 58.8 | 59.8 | 43.5 | 46.4 | 59.1 |
| LLaVA w/ Multilingual data | 67.0 | 59.1 | 60.3 | 44.2 | 48.1 | 59.7 |
| PARROT | 70.0 | 68.1 | 67.3 | 62.7 | 58.0 | 66.3 |

*Table 10.* The comparison of the translation-based baseline and PARROT. While the naive baseline shows some improvements in certain languages, such as Chinese, it leads to performance degradation in others, such as Russian and Portuguese.

| Method | MMMB | | | | | |
|---|---|---|---|---|---|---|
| | *en* | *zh* | *pt* | *ar* | *tr* | *ru* |
| LLaVA | 67.1 | 58.8 | 59.8 | 43.5 | 46.4 | 59.1 |
| LLaVA w/ translation | 67.1 | 60.7 | 58.6 | 47.3 | 48.6 | 58.9 |
| PARROT | 70.0 | 68.1 | 67.3 | 62.7 | 58.0 | 66.3 |

increase the multilingual data, the model's performance on the MMMB benchmark continues to improve, suggesting that our model can handle imbalanced multilingual data while still achieving effective scaling and performance gains.

*Table 11.* The performance comparison on MMMB when scaling the sample sizes of **each language**.

| Sample Size | MMMB | | | | | |
|---|---|---|---|---|---|---|
| | *en* | *zh* | *pt* | *ar* | *tr* | *ru* |
| 10K | 70.0 | 68.1 | 67.3 | 62.7 | 58.0 | 66.3 |
| 30K | 70.1 | 68.0 | 67.6 | 64.1 | 59.9 | 66.7 |
| 50K | 69.9 | 67.9 | 67.8 | 64.8 | 61.4 | 67.2 |
| 70K | 70.3 | 68.4 | 68.3 | 65.7 | 63.2 | 67.4 |

*Table 12.* The performance comparison on MMMB when scaling model sizes of Qwen-series LLM.

| Method | MMMB | | | | | |
|---|---|---|---|---|---|---|
| | *en* | *zh* | *pt* | *ar* | *tr* | *ru* |
| PARROT-7B | 70.0 | 68.1 | 67.3 | 62.7 | 58.0 | 66.3 |
| PARROT-14B | 73.9 | 71.6 | 69.8 | 68.1 | 64.3 | 70.1 |
| PARROT-32B | 76.3 | 75.4 | 73.8 | 72.1 | 71.2 | 73.5 |

Additionally, we extend PARROT's LLM backbone from Qwen1.5-7B to Qwen1.5-32B, using the same model design and configuration, and evaluate them on the MMMB dataset. As shown in Table 12, the results indicate that PARROT continues to yield better performance even with a larger LLM backbone. This finding validates the idea that the scaling law for model parameters still holds, and our design remains effective as the model size increases. While we are currently limited to the Qwen1.5-32B model, these results suggest that our approach can scale well with model size, and we believe similar trends would be observed with even larger models, such as those with 30B parameters or beyond.

# D. More Implementation Details

## D.1. MoE Training Strategy

During the first pre-training stage, the MoE module is initialized with random parameters but is not activated or included in the training process. Instead, we focus exclusively on training the projector. This avoids the issue of training a good projector under a randomly initialized MoE. In detail:

**1) Pre-training Stage.** In this stage, the MoE module is bypassed entirely, meaning the image tokens do not pass through the MoE. The primary goal of this stage is to train the projector using a large number of image-text pairs. This enables the projector to align image tokens and textual tokens effectively without interference from the untrained MoE module.

**2) SFT Stage.** Since the SFT stage requires the participation of MoE modules, we randomly initialize the parameters of the MoE components prior to the SFT phase. Once the projector has been trained and achieves robust alignment capabilities in the pre-training stage, we introduce multilingual training data and activate the MoE parameters. At this stage, the MoE is optimized with textual guidance, which drives the alignment of visual tokens while leveraging the well-trained projector. The prior alignment achieved in the pre-training stage allows the MoE to optimize efficiently during this phase.

We present the entire training process of PARROT in the form of pseudocode, as shown in Algorithm 1. It is clear from the algorithm that during the pre-training phase, only the projector is trained. Before the start of the SFT phase, the MoE modules are randomly initialized and incorporated into the training process during the SFT phase.

---

**Algorithm 1** PARROT for multilingual MLLM

---

**Input**: Pre-training datasets: $\mathcal{D}^1$, SFT datasets: $\mathcal{D}^2$;

 1: Construct the training data in LLaVA format;
 2: Activate the parameters of the projector and freeze others;
 3: **for** each data in $\mathcal{D}^1$ **do**                                               ▷ Pre-training stage
 4:      Optimize the projector to effectively bridge the modality gap;
 5: **end for**
 6: Randomly initialize the parameters of MoE.
 7: Activate the parameters of the projector, LLM, and MoE;
 8: **for** each data in $\mathcal{D}^2$ **do**                                                    ▷ SFT stage
 9:      Select the multilingual experts based on the textual guidance;
10:      Optimize the projector, LLM, and MoE;
11: **end for**

---

*Table 13.* The detailed training hyperparameters.

| Config | Stage 1 | Stage 2 |
|---|---|---|
| Experts | - | 6 |
| MLP expert network | 2 Linear layers with SiLU | |
| Deepspeed | Zero2 | Zero3 |
| Image resolution | 336×336 | |
| Image encoder | Clip-ViT-L/14-336 | |
| Feature select layer | -2 | |
| Image projector | 2 Linear layers with GeLU | |
| Epoch | 1 | |
| Optimizer | AdamW | |
| Learning rate | 1e-3 | 2e-5 |
| Learning rate scheduler | Cosine | |
| Weight decay | 0.0 | |
| Text max length | 2048 | |
| Batch size per GPU | 16 | 8 |
| GPU | 16 × A100-80G | |
| Precision | Bf16 | |
| Gradient checkpoint | True | |

## D.2. More Experimental Details about Different Backbones

In the following, we provide detailed information to explain Figure 1. Firstly, to ensure a fair comparison between the OpenAI-CLIP-based model and the Chinese-CLIP-based model, we train distinct models using the same training data as LLaVA, as shown in Table 2a. The hyperparameters are listed in Table 13 without the MoE hyperparameters. As depicted in Figure 1, the OpenAI-CLIP-based model struggles to generate Chinese outputs when given Chinese prompts due to the English-centric training data. In contrast, despite the extremely scarce amount of Chinese training data, the Chinese-CLIP-based model naturally acquires zero-shot capability to understand, process, and generate Chinese texts. Furthermore, we compare both models on MMBench-CN and MMMB-zh to evaluate their Chinese capability. As shown in Table 8, the performance of the Chinese-CLIP-based model is significantly higher than that of the OpenAI-CLIP-based model. On the other hand, we empirically find that different LLMs have a significant impact on performance. Qwen (Bai et al., 2023a) demonstrates superior Chinese capability compared to Vicuna (Chiang et al., 2023), yet its English capability remains competitive.

## D.3. Implementation Details of PARROT

As shown in Table 13, we provide the training hyperparameters for PARROT. Throughout all stages of training, we consistently train for one epoch, with a batch size of 256 for the first stage and 128 for the second stage. We maintain an image resolution of 336x336 for all two stages and enable the gradient checkpoint mode for each training stage.

### D.4. Analysis of the Translation-based Baseline

There is a naive baseline where we first translate the question into English and then translate the English answer back to the target language. On the one hand, our experimental setting follows recent work in multilingual and multimodal large language models (Hu et al., 2023; Zhang et al., 2024a; Hinck et al., 2024), where such a naive baseline has not been commonly considered. While the translation-based approach could be a straightforward alternative, it faces some significant challenges.

First, it is highly susceptible to translation noise, particularly issues related to polysemy and meaning ambiguity between languages. Moreover, our benchmark includes a substantial number of cultural-specific questions, which require deep cultural context knowledge that translation alone cannot effectively capture. In practical use, adding an additional translation step would also introduce extra overhead, increasing both the time and computational cost.

Despite these challenges, we acknowledge the importance of evaluating this baseline and conducting experiments to assess the performance of this translation-based baseline by using the Google Translation API. As shown in the Table 10, the results reveal a "seesaw effect"—while the naive baseline shows some improvements in certain languages, such as Chinese, it leads to performance degradation in others, such as Russian and Portuguese. This highlights the difficulty of addressing multilingualism and multimodal tasks solely through translation.

## E. Discussion

PARROT is a novel approach that leverages textual guidance to align visual tokens at the language level, enabling the conversion of English-biased visual embeddings into language-specific ones through an MoE module. In future work, we plan to incorporate more culture-related samples in various languages. This will enhance the representation of diverse cultural contexts and ensure that our benchmark accurately reflects the complexities of multilingual interactions. Additionally, we will focus on developing tasks that not only assess linguistic capabilities but also evaluate cultural nuances, which are crucial for effective communication in multilingual settings. By doing so, we aim to provide a more comprehensive evaluation of multilingual models and their performance across different cultural backgrounds.

## F. More Visualization Results

In this section, we include additional visualization results between users' questions and PARROT's responses using multiple languages. These pictures are selected from LLaVA (Liu et al., 2023b) and CuMo (Li et al., 2024c). As depicted in Figures Figures 12 to 17, it is evident that PARROTpossesses superior multilingual capabilities for understanding, processing, and generating multilingual texts. In certain specific cases, PARROT may also experience hallucinations. As depicted in the upper case of Figure 12, it misidentifies Xiaomi SU7 as a Porsche Taycan.

*Table 14.* **Ablation study on monolingual fine-tuning dataset in MMMB benchmark.** The table shows an effect of performance on six languages when using fine-tuning data from different languages. Models with 7B parameters are used for this ablation.

| Dataset | English | Chinese | Portuguese | Arabic | Turkish | Russian |
|---|---|---|---|---|---|---|
| LLaVA-1.5-finetune | **72.69** | 67.60 | 65.61 | 57.72 | 48.30 | 63.80 |
| + *zh* 71k | 69.18 | **69.06** | 63.92 | 58.13 | 48.95 | 63.63 |
| + *pt* 14k | 69.94 | 68.83 | 65.67 | 58.65 | 51.11 | 63.04 |
| + *ar* 12k | 70.47 | 68.36 | 64.39 | 60.79 | 51.11 | 63.16 |
| + *tr* 17k | 70.82 | 69.01 | 64.85 | 60.76 | **60.70** | 64.39 |
| + *ru* 14k | 69.59 | 68.07 | 64.27 | 60.35 | 53.92 | 64.15 |
| + *zh pt ar tr ru* | 70.00 | 68.13 | **67.31** | **62.69** | 58.01 | **66.26** |

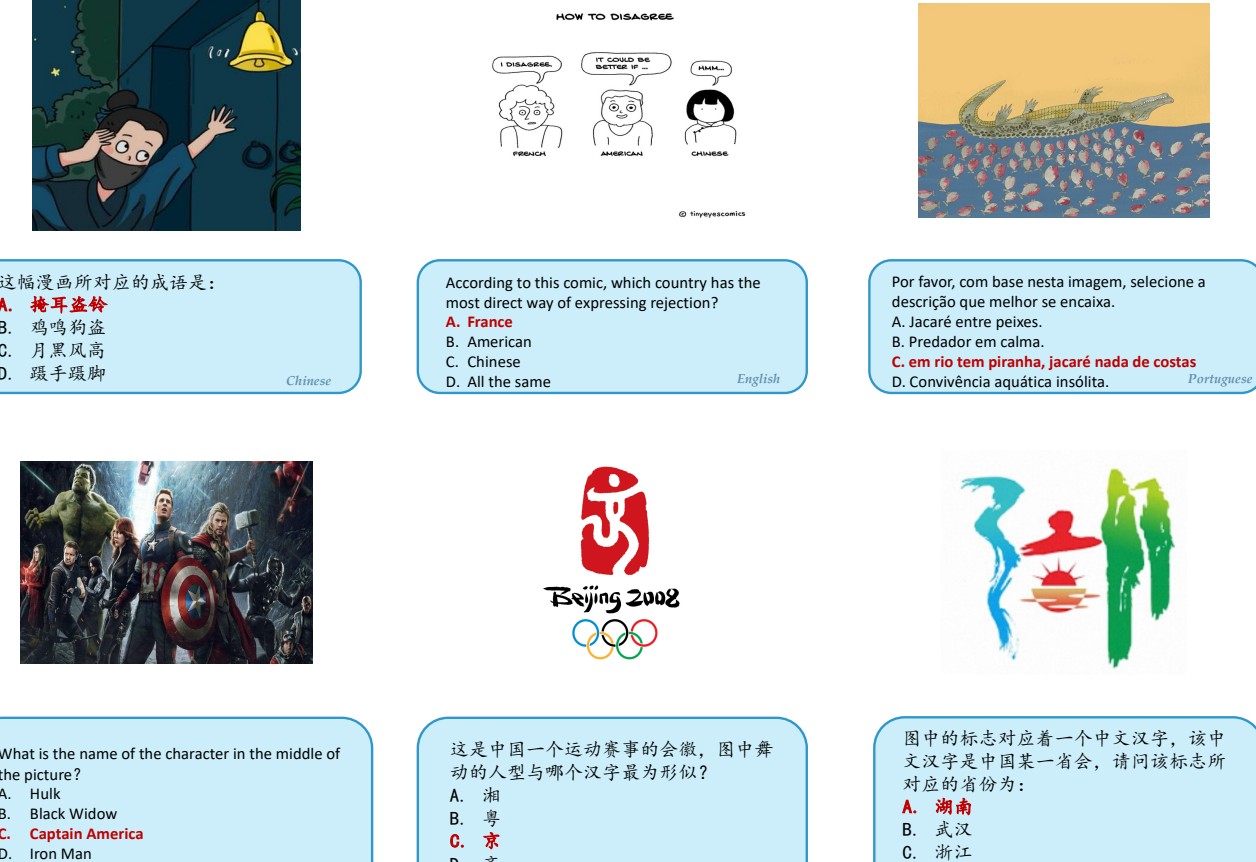

*Figure 11.* Several culture-related samples in different languages.

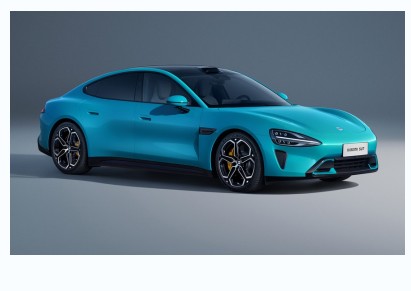

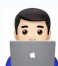 **English**: What car is it in this image?

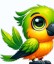 Output: The turquoise blue car is Porsche Taycan.

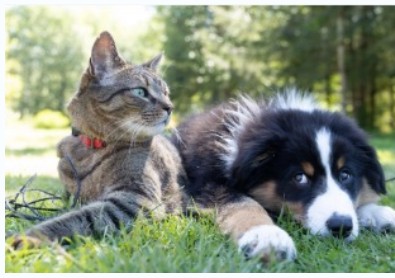

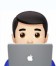 **English**: What animals are in this image?

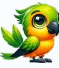 Output: A cat and a dog are in the image.

*Figure 12.* More visualization results between the user and PARROT using English prompts. We highlight the hallucinations from the responses of PARROT.

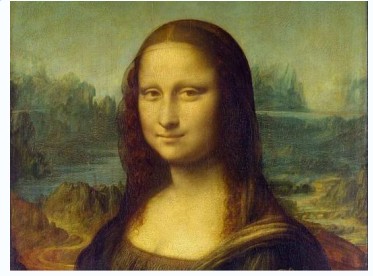

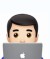 **Chinese**: 你知道这幅画是谁画的吗?
[Chinese-English Trans: Do you know who drew this painting]

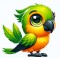 Output: 莱昂纳多·达芬奇
[Chinese-English Trans: Leonardo da Vinci]

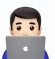 **Chinese**: 请描述这幅画的主要内容
[Chinese-English Trans: Please describe the main content of this painting.]

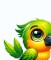 Output: 这幅画是一幅肖像画，描绘了一个年轻女子的头部和肩膀。她穿着一件黑色连衣裙，上面有金色的装饰。她的头发是深棕色的，被梳成一个发髻。她的脸上有褐色的斑点，眼睛是棕色的。她正在微笑，表明她心情愉快。这幅画的风格是肖像画，它以描绘人物的头部和肩膀为特征。肖像画通常使用逼真的技巧，以捕捉人物的特征和表情。这幅画中人物的服装和环境暗示着它可能描绘的是文艺复兴时期的艺术家。
[Chinese-English Trans: This painting is a portrait, depicting the head and shoulders of a young woman. She is dressed in a black dress with gold embellishments. Her hair is dark brown and styled into a bun. There are brown spots on her face, and her eyes are brown. She is smiling, indicating she is in a good mood. The style of the painting is portraiture, characterized by the depiction of the subject's head and shoulders. Portraits often use realistic techniques to capture the features and expressions of the person. The attire and setting of the figure in this painting suggest it may be depicting an artist from the Renaissance period.]

*Figure 13.* More visualization results between the user and PARROT using Chinese prompts.

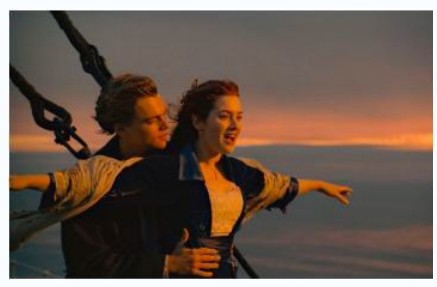 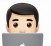

**Portuguese**: Qual é o final deste filme?
[Portuguese-English Trans: What's the ending of this movie?]

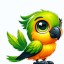

Output: O filme "Titanic" é um romance histórico de 1997 que apresenta uma narrativa fictícia de uma mulher chamada Rose e um homem chamado Jack. A narrativa se passa durante o final do século XIX, quando o navio "Titanic" está se preparando para sua viagem de luxo para Nova York. Jack e Rose se enamoram e se casam, mas a viagem é marcada por um desastre inesperado.
[Portuguese-English Trans: The movie "Titanic" is a 1997 historical romance that features a fictional narrative about a woman named Rose and a man named Jack. The story is set during the late 19th century, as the ship "Titanic" is preparing for its luxury voyage to New York. Jack and Rose fall in love and get married, but the journey is marked by an unexpected disaster.]

*Figure 14.* More visualization results between the user and PARROT using Portuguese prompts.

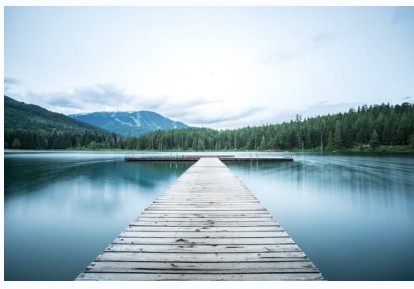 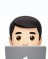

**Arabic**: ما الأشياء التي يجب أن أكون حذرًا منها عند زيارتي هنا؟
[Arabic-English Trans: What's the ending of this movie?]

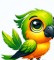

Output: أثناء زيارة هذا الموقع، يجب أن يكون حذرًا من أي مخاطر محتملة، مثل الأسطح المبللة، أو الأشياء المائية، أو أي عوائق محتملة، مثل الصخور أو الصخور أو الأشياء المائية. بالإضافة إلى ذلك، يجب على الزوار أن يكونوا على علم بالعوامل البيئية، مثل تغيرات الطقس، أو درجات الحرارة، أو الظروف المائية، لضمان تجربة آمنة ومرضية.
[Arabic-English Trans: During the visit to this site, one should be cautious of any potential hazards, such as wet surfaces, or water-related objects, or any potential obstacles, like rocks or boulders or aquatic objects. Additionally, visitors should be aware of environmental factors, such as changes in weather, temperature, or water conditions, to ensure a safe and satisfying experience.]

*Figure 15.* More visualization results between the user and PARROT using Arabic prompts.

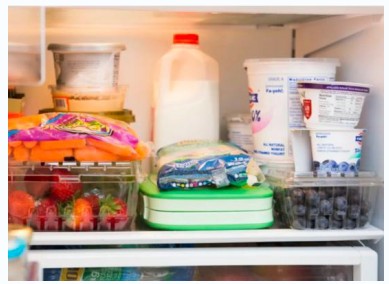
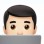

**Turkish**: Bu malzemelerle hangi yemekleri yapabilirim?
[Turkish-English Trans: What are the meals that I can cook with these?]

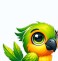

Output: Buzdolabındaki çeşitli malzemelerle, çilekli bir salata, havuçlu bir sandviç, meyveli bir yemek ve süt ve yoğurtla bir yemek gibi çeşitli yemekler yapılabilir. Buzdolabı, havuç, çilek ve yoğurt gibi diğer gıda maddelerini de içerir; bunlar çeşitli yemekler yaratmak için kullanılabilir. Birden fazla yiyecek maddesi ve kapların olması, buzdolabının iyi stoklandığını ve çeşitli yemek seçeneklerine olanak tanıdığını gösterir.
[Turkish-English Trans: With the various items in the refrigerator, one can cook a variety of meals, such as a salad with strawberries, a sandwich with carrots, a meal with berries, and a meal with milk and yogurt. The refrigerator also contains other food items, such as carrots, strawberries, and yogurt, which can be used to create a diverse range of dishes. The presence of multiple food items and containers suggests that the refrigerator is well-stocked, allowing for a variety of meal options.]

*Figure 16.* More visualization results between the user and PARROT using Turkish prompts.

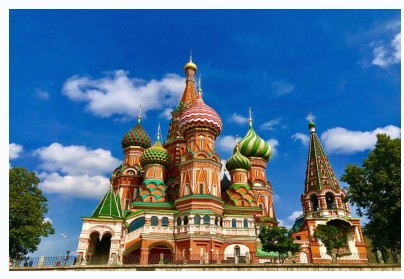
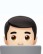

**Russian**: Пожалуйста, помогите мне представить это место на фотографии.
[Russian-English Trans: Please help me introduce the place in this picture.]

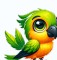

Output: Место, изображенное на картинке, - это собор Василия Блаженного, официально известный как собор Покрова Пресвятой Богородицы на рву. Это знаковое строение находится в Москве, Россия, на Красной площади.
[Russian-English Trans: The place depicted in the picture is the Saint Basil's Cathedral, officially known as the Cathedral of the Intercession of the Most Holy Theotokos on the Moat. This iconic structure is located in Moscow, Russia, on Red Square. ]

*Figure 17.* More visualization results between the user and PARROT using Russian prompts.

