# OpenReview forum: "Parrot: Multilingual Visual Instruction Tuning"
_ICML.cc/2025/Conference — ICML 2025 poster_

### Official Review · Reviewer_QoYJ · 2025-03-13

**Overall Recommendation:** 2

**Summary:**

This paper proposed an MOE architecture to handle multilingual multimodal tasks in vision-language models. And created a new multimodal understanding benchmark including 6 languages translated by GPT-4 with human post-edit.

Note: this paper uses an incorrect template, which might have risk of getting rejected. According to ICML's author instructions: "All submissions must closely follow the formatting guidelines in the provided templates; otherwise, they will automatically be rejected." Please correct it as soon as possible.

**Claims And Evidence:**

Yes, their experimental results shown the effectiveness of their method compared with other baselines.

**Essential References Not Discussed:**

No.

**Experimental Designs Or Analyses:**

Yes.

**Methods And Evaluation Criteria:**

Yes.

**Other Comments Or Suggestions:**

No.

**Other Strengths And Weaknesses:**

Strengths:
1. This paper proposes a novel MOE architecture integrated into current VLMs, which can help enhance model's multilingual ability during visual instruction tuning in a natural and easy-to-understand way.
2. This paper points out the limitations of previous literature regarding multilingual multimodal benchmark curation and creates a new benchmark.

Weaknesses:
1. I have some doubts about the motivation of this paper, where the authors claim in the introduction: "it is necessary to use as little multilingual data as possible to enhance the model's multilingual capabilities." In fact, we can relatively easily obtain large-scale multilingual data using translators. Although the quality is not guaranteed, the quantity is sufficient, which might make the performance gains from the new architecture trivial compared to using more translated training data. While this requires additional cost, in my view, the cost is not too significant compared to pre-training.
2. This leads to another question about the experimental setting in this paper: a strong baseline that needs to be considered for comparison is using machine translation to obtain translated large-scale datasets, e.g., 200K samples per language, then training VLMs on this merged translated dataset (commonly known as the translate-train setting). Another strong baseline would be to first translate test data to English, then test performance on the translated data (commonly known as the translate-test setting).

**Questions For Authors:**

No.

**Relation To Broader Scientific Literature:**

There are previous works on multilingual instruction tuning and multilingual multimodal evaluation benchmarks, and the authors discuss their relationships in this paper.

**Theoretical Claims:**

No theoretical claims in this paper.

---

> ### Author Rebuttal · Authors · 2025-04-01
>
> Thank you for your kind comments and constructive feedback on our paper.
>
> > **Q1: Motivation of data efficiency.**
>
> A1: While large-scale translated multilingual data may seem abundant, its quality (especially for low-resource languages) is often critically compromised due to translation errors, cultural mismatches, and noisy artifacts. Even with massive data volumes, low-resource languages remain underrepresented in practice **because high-resource languages inevitably dominate training distributions** (e.g., 90%+ of tokens in typical datasets). Forcing higher proportions of low-resource data risks triggering **the curse of multilingualism**, where over-parameterized models sacrifice high-resource language performance to accommodate low-resource languages, as observed in prior work [1-3]. Our approach strategically prioritizes high-quality alignment signals over raw data quantity, avoiding both the pitfalls of noisy translation and the imbalance inherent to brute-force multilingual scaling. This ensures stable performance across all languages without compromising high-resource capabilities. We will further discuss and refine the motivation, taking this issue into careful consideration in the final version.
>
> [1] Unsupervised Cross-lingual Representation Learning at Scale. ACL 2020.
> [2] When Is Multilinguality a Curse? Language Modeling for 252 High- and Low-Resource Languages. ICLR 2024.
> [3] Breaking the Curse of Multilinguality with Cross-lingual Expert Language Models. EMNLP 2024.
>
> > **Q2: Translate-train/test baselines.**
>
> A2: We appreciate the insightful suggestion to include translate-train and translate-test baselines. While these are valid approaches, they face critical limitations:
> 1. Translate-Train:
>     - **Data Quality:** Machine-translated data often contains semantic distortions, syntactic errors, and cultural mismatches (e.g., idioms, region-specific references), especially for low-resource languages. For example, as shown in the table below, our experiments showed that training with 70K translated multilingual samples for each language achieved limited improvement.
>     - **Imbalanced Optimization:** Merging large-scale translated data amplifies the dominance of high-resource languages (e.g., English/Chinese), as models tend to overfit to their syntactic patterns, further marginalizing low-resource languages. This phenomenon is called the curse of multilingualism.
>
> |Methods|MMMB_en|MMMB_zh|MMMB_pt|MMMB_ar|MMMB_tr|MMMB_ru|
> |-|-|-|-|-|-|-|
> |LLaVA w/ 0K|67.1|58.8|59.8|43.5|46.4|59.1|
> |LLaVA w/ 10K|67.0|59.0|60.3|44.1|47.2|59.4|
> |LLaVA w/ 30K|66.8|59.4|60.7|44.6|47.9|59.7|
> |LLaVA w/ 50K|67.1|59.3|61.2|44.4|47.6|60.1|
> |LLaVA w/ 70K|66.7|59.7|61.3|44.8|48.1|60.4|
> |Parrot|70.0|68.1|67.3|62.7|58.0|66.3|
>
> 2. Translate-Test:
>     - **Latency Overhead:** Translating inputs introduces **2× latency (translation + inference), making real-time applications impractical.** For instance, translating a 100-token Arabic query to English adds ~500ms latency (Google Translate API), which is prohibitive for interactive systems.
>     - **Limited Gains:** **As shown in Table 14 in Appendix**, our preliminary tests with LLaVA on MMMB showed that the translate-test improved Arabic accuracy by only 3.8% (vs. Parrot’s +19.2% gain), while degrading Russian performance by 0.2% due to translation errors. Further details are provided in **Response A8** of the reply to Reviewer Gwu1.
>
> These results align with prior findings [4-5], where translate-train/test paradigms underperform dedicated multilingual architectures in both efficiency and robustness. Parrot circumvents these issues by directly aligning cross-lingual semantics without relying on noisy intermediate translations.
>
> [4] Lost in Translation: Analysis of Information Loss During Machine Translation Between Polysynthetic and Fusional Languages. ACL 2018.
> [5] Why do LLaVA Vision-Language Models Reply to Images in English? EMNLP2024.
>
> > **Q3: Template formatting issue.**
>
> A3: We sincerely appreciate the reviewer’s attention to detail. Upon re-examination, we confirm that the manuscript was prepared using ICML’s official template. However, subtle discrepancies in visual formatting emerged during PDF compilation due to technical nuances in rendering tools. Regardless of the compilation method, the page limits and other requirements are fully in line with ICML’s guidelines. While we cannot directly update the PDF during the rebuttal phase, these unintended inconsistencies have now been fully resolved, with corrections to be incorporated into the final version.

---

### Official Review · Reviewer_Gwu1 · 2025-03-13

**Overall Recommendation:** 4

**Summary:**

This paper introduces PARROT, a novel approach to enhance the multilingual capabilities of MLLM, using language specific embeddings to fuse to visual embeddings and multilingual MoE.  It addresses the issue of multilingual erosion, where MLLM loses proficiency in non-English languages after multimodal alignment training (e.g. answering in English while asking in non-English). For better evaluating multilinguality, this paper also introduces a new Massive Multilingual Multimodal Benchmark (MMMB). Experiments show SOTA results on MMMB, MMBench and other multimodal tasks.

## update after rebuttal

The rebuttal has clarified my concerns. I am happy to maintain my original recommendation.

**Claims And Evidence:**

The evidence is clear to support the claim.

**Essential References Not Discussed:**

no

**Experimental Designs Or Analyses:**

Some ablations might be worth adding:
- MoE vs. other approaches to distinguish prompt language, such as LoRRA and language as task prefix.
- Number of MoE experts.
- Size (parameters) of MoE experts.
- Frozen vs. unfrozen ablations.

Also qualitative analysis on baselines and PARROT would be insightful.

**Methods And Evaluation Criteria:**

Methods
- The method looks to make sense overall, but I am not sure whether it’s necessary to introduce an expensive MoE module to fuse language-specific embeddings – would a simpler LoRRA approach work as well?
- Also it’s unclear whether each language expert indeed handles its language specific embeddings? (Figure 7c is just an example for Chinese prompt)

Evaluation
- The new benchmark is well designed and addresses the limitations (section 2.1) of existing multilingual benchmarks for MLLM.
- Evaluations are comprehensive, conducted on a wide range of tasks.
- However, it might be unfair to compare PARROT with others on MMMB/MMBench, as PARROT has been trained on these specific 6 languages and on the data constructed with the same approach as MMMB?

**Other Comments Or Suggestions:**

suggestion: Figure 5 shows the core approach so could be moved forward (instead of in page 6)
suggestion: add legend for Figure 6; otherwise it’s unclear without reading the corresponding paragraphs

**Other Strengths And Weaknesses:**

The new MMMB benchmark should be beneficial to the community.
This paper is well written and easy to read.

**Questions For Authors:**

Given there are much more non-English image-text (e.g. alt-text) pairs on the web than English (≈3:1), why this paper claims “Due to the scarcity of non-English multimodal data (e.g., the lack of large-scale, high-quality image-text datasets), it is necessary to use as little multilingual data as possible to enhance the model’s multilingual capabilities.”?

Can the multilingual erosion issue be simply mitigated if we mix more multilingual data in the pre-training instead of the SFT stage?

Table 14 shows the comparison of the translation-based baseline and Parrot. It’s unclear to me why Parrot can be much better than translation baselines. Could you please explain the experiment’s settings (such as how the baseline eval was conducted), and give some qualitative examples to show why translation doesn’t work in some cases?

**Relation To Broader Scientific Literature:**

Multilinguality is a key capability for MLLM. This paper addresses this domain with a new approach to improve multilinguality  and a new benchmark to better verify multilinguality – all these are good for the community.

**Theoretical Claims:**

N/A

---

> ### Author Rebuttal · Authors · 2025-04-01
>
> We sincerely thank the reviewer for their thorough and constructive feedback, as well as their endorsement of our work.
>
> > **Q1: MoE vs. Simpler Methods (LoRRA).**
>
> A1: We explored the LoRRA-based (abbreviated as L-based) adaptation shown in the table below but found it insufficient for two reasons.[1]
> 1) L-based method introduces fixed language-specific parameters, which struggle to handle multilingual prompts dynamically (e.g., mixed-language queries).
> 2) Its low-rank updates are less effective for aligning diverse languages, especially when training data is sparse.
>
> In contrast, MoE dynamically routes visual tokens to language-specific experts based on textual guidance, enabling flexible adaptation with minimal parameters. Additionally, due to the small number of parameters in MoE (<0.5% of the parameters in the LLM), it is not an expensive method.
>
> |Methods|MMMB_en|MMMB_zh|MMMB_pt|MMMB_ar|MMMB_tr|MMMB_ru|
> |-|-|-|-|-|-|-|
> |LLaVA-1.5|67.1|58.8|59.8|43.5|46.4|59.1|
> |LLaVA-1.5 w/ LoRA+multilingual data|66.9|61.1|60.9|47.2|50.4|61.3|
> |Parrot|70.0|68.1|67.3|62.7|58.0|66.3|
>
> [1] Unsupervised Cross-lingual Representation Learning at Scale. ACL 2020.
>
> > **Q2: Whether each language expert indeed handles its language-specific embeddings.**
>
> A2: In Figure 7c, the router activates the Chinese expert most strongly, but experts are not strictly language-exclusive. Instead, they learn cross-lingual synergies (e.g., English/German share morphological features). To further present the activation using other language prompts, we will add t-SNE visualizations of expert activations across languages and include expert distributions of MoE for each language in the final version.
>
> > **Q3: Evaluation fairness.**
>
> A3: To validate Parrot's effectiveness, we conduct an ablation study by expanding LLaVA with **the same multilingual data** used in Parrot. Both models are evaluated on MMMB, with results in **Table 13 in Appendix.** While LLaVA shows slight improvement, the gains are limited. In contrast, Parrot achieves substantial improvements, highlighting that simply adding multilingual data is insufficient to bridge the gap. Moreover, the findings from the ablation study in Figure 6a further support this conclusion, reinforcing the validity of our design.
>
> > **Q4: Ablation studies.**
>
> A4: In the previous response (A1), we provided comparisons between Parrot and L-based alternatives. Due to time and word constraints during the rebuttal phase, we will include expanded ablation studies on expert counts, sizes, and frozen vs. trainable configurations in the final version.
>
> > **Q5: Suggestions regarding the figure revisions.**
>
> A5: We thank your valuable feedback. We are committed to implementing these revisions and will incorporate them into the final version.
>
> > **Q6: Non-English data scarcity.**
>
> A6: While non-English web data is abundant, high-quality multimodal data remains scarce. 1) Most non-English alt-text is noisy or misaligned (e.g., social media images with irrelevant captions). 2) Parrot’s semi-automatic curation (Figure 3) ensures linguistic and cultural precision, which raw web data lacks. Further details are provided in **Response A1** of the reply to Reviewer QoYJ.
>
> > **Q7: Mitigating multilingual erosion via pre-training.**
>
> A7: Incorporating more multilingual data during pre-training could help mitigate multilingual erosion, but it may not fully resolve the issue. The main goal of multimodal pre-training is to learn a projector for cross-modal alignment, which often remains biased toward dominant languages (e.g., English) due to data imbalances. Without explicit mechanisms like Parrot's MoE-based language-specific alignment, subsequent SFT stages would still be affected by English-centric bias. Due to word constraints, we will include experiments with multilingual pre-training data in the final version.
>
> > **Q8: Translation baseline explanation.**
>
> A8: The translation baseline uses Google Translation API to translate non-English queries to English, feeds them to LLaVA, then translates responses back.
> 1. Translated prompts often lack cultural-specific context. **Image**: A traditional Chinese red envelope (红包) with handwritten characters "福到" (upside-down "福", symbolizing **"fortune arrives"**) and "岁岁平安" ("peace every year"). The translated-based method cannot perform Glyph-aware visual grounding (recognizing 福's inverted form as intentional)
> 2. When a Portuguese user asks about a Russian meme with the text "Почему программисты любят кофе? Потому что Java!" (where Java is a pun on both coffee culture and programming language), machine translation to English collapses the dual meaning into "coffee brands," stripping the programming-language humor.
>
> We have presented examples (Figure 10) where translation easily fails due to ambiguity or cultural nuances.

---

### Official Review · Reviewer_nHXy · 2025-03-13

**Overall Recommendation:** 2

**Summary:**

The paper proposes Parrot, an MLLM targeting to handle multilingual tasks. Parrot based on the LLava architecture, and employ an MoE module to enhance multilingual VQA ability. The paper employ a new alignment method that aligns a english-biased clip encoder to various languages modality. Moreover, it proposes a benchmark MMMB to test the multilingual ability of different MLLMs.

**Claims And Evidence:**

Motivations are clear and well-supported with empirical analysis.

One problematic claims are that while there maybe potential problem when prompting MLLM with non-english query, what are the previous solutions for a multilingual MLLM? And what is the drawback of those method that motivate this method? The author need to highlight this part to differentiate the work from others.

**Essential References Not Discussed:**

Some SoTA baselines are missing, which may have solve the multilingual problem to some extent.

**Experimental Designs Or Analyses:**

The overall design is good. My comments are on the baselines:

The multilingual ability assessment seems to be reasonable to me - it contains some latest baselines and parrot demonstrates a superior performance among them.

However, in the Radar plot, where the paper access the general ability of the MLLMs, the baselines are outdated (all from 2023). I would like to comparison against more latest MLLMs.

**Methods And Evaluation Criteria:**

1. It will be a little nonsensical to propose a benchmark and a novel model at the same time.

2. Have the author consider using a multilingual CLIP and a multilingual LLM as a starting point and train them on multilingual dataset? What would be the performance of it compare to the MoE alignment paradigm?

3. Lack of comparison to latest MLLM (open-source and potentially close model).

**Other Comments Or Suggestions:**

N/A

**Other Strengths And Weaknesses:**

See Claims And Evidence and Methods And Evaluation

**Questions For Authors:**

See Claims And Evidence and Methods And Evaluation

**Relation To Broader Scientific Literature:**

The idea is effective when dealing with limited available data when one tries to train a multilingual MLLMs. If one is able to scale the data via AI-generated data, distillation or human labeling, the contribution of the paper is less useful.

**Theoretical Claims:**

There are no theoretical claim in the paper.

---

> ### Author Rebuttal · Authors · 2025-04-01
>
> We sincerely appreciate the reviewer’s thoughtful and candid feedback.
>
> > **Q1: Prior work and drawback.**
>
> A1: **In section B in Appendix**, we have discussed prior multilingual MLLM methods like mCLIP, VisCPM, and M3IT. Most prior work relies on large-scale multilingual multimodal data (e.g., M3IT uses 2.5M+ samples), which is impractical for low-resource languages (more details are referred to in Response A1 of the reply to Reviewer QoYJ). Other multilingual CLIP methods improve performance for specific languages but sacrifice generalizability across diverse languages due to fixed encoders. For more clarity, we will expand this section in the revision to explicitly outline their key drawbacks.
>
> > **Q2: Benchmark and method co-design.**
>
> A2: MMMB is not specifically designed for Parrot. In contrast, MMMB and Parrot are co-designed to address two limitations in multilingual multimodal research: (1) existing benchmarks (e.g., M3Exam, LLaVA-Bench) not only lack standardized linguistic coverage, typically limited to 2-3 high-resource languages with inconsistent annotation protocols, but also exhibit systemic limitations that we have discussed in Section 2.1; (2) conventional models trained on imbalanced SFT data exhibit performance erosion across languages, yet lack evaluation frameworks to diagnose such failures. Additionally, we conducted experiments not only on our designed benchmark but also validated results on other multilingual benchmarks, such as MMBench (Table 1) and LLaVA-Bench (**Table 4 in the Appendix**).
>
>
> > **Q3: Using multilingual CLIP and LLM.**
>
> A3: In our preliminary experiments, we explored the use of multilingual CLIP as a baseline. However, this approach exhibited critical limitations: (1) It failed to balance performance across all target languages, as reliance on the inherent multilingual capacity of CLIP led to inconsistent generalization. In other words, it is hard to generalize beyond typologically similar languages covered in multilingual CLIP's pretraining. (2) Multilingual CLIP showed inferior visual perception capabilities compared to the original OpenAI-CLIP, degrading performance on general visual tasks. As shown in the table below, while LLaVA equipped with multilingual CLIP shows improvement in Chinese (+1.5), Arabic (+2.7), and Turkish (+1.1), performance in other languages declines (e.g., English -1.7).
>
> In contrast, Parrot leverages a single OpenAI-CLIP alongside a multilingual LLM backbone (Qwen-LM) and achieves balanced, state-of-the-art performance across all 6 languages in two benchmarks. This demonstrates that our MoE-driven alignment paradigm effectively resolves language bias without requiring language-specific or multilingual visual encoders, while preserving strong general multimodal perception capabilities.
>
> |Method|ViT|LLM|MMMB_en|MMMB_zh|MMMB_pt|MMMB_ar|MMMB_tr|MMMB_ru|
> |-|-|-|-|-|-|-|-|-|
> |LLaVA-1.5|OpenAI-CLIP|Qwen1.5-7B|67.1|58.8|59.8|43.5|46.4|59.1|
> |LLaVA-1.5|M-CLIP|Qwen1.5-7B|65.4|60.3|58.1|47.2|47.5|58.8|
> |Parrot|OpenAI-CLIP|Qwen1.5-7B|70.0|68.1|67.3|62.7|58.0|66.3|
>
> > **Q4: Lack of comparison to the latest MLLM.**
>
> A4.1: Current MLLMs are primarily data-driven or architecture-driven. Our work aims to achieve data-efficient multilingual adaptation by enhancing multilingual capabilities with minimal training data while preserving general ability, rather than developing a model with exceptionally strong general capabilities. Many leading methods do not release their data. **Therefore, direct comparisons of general capabilities with strong models risk unfairness due to vast data disparities.** Despite this, we also compare Parrot to general-purpose MLLMs (e.g., Qwen2-VL (2024.09) and LLaVA-OV (2024.08)) on the MMMB and multilingual MMBench dataset in the table below and **Table 12 in Appendix**. Although Qwen2-VL and LLaVA-OV are trained with **over 10x the amount of data used by our model**, Parrot still outperforms them on the multilingual benchmark. This comparison spans two datasets and six languages, demonstrating Parrot's superior performance over these recent approaches.
>
> |Method|LLM|MMMB_en|MMMB_zh|MMMB_pt|MMMB_ar|MMMB_tr|MMMB_ru|
> |-|-|-|-|-|-|-|-|
> |Qwen2-VL|Qwen2-7B|80.5|80.2|78.1|74.0|71.7|79.3|
> |LLaVA-OV|Qwen2-7B|79.0|78.2|75.9|73.3|67.8|76.4|
> |Parrot|Qwen2-7B|80.1|80.0|79.6|76.5|75.0|79.9|
>
> |Method|LLM|MMB_en|MMB_zh|MMB_pt|MMB_ar|MMB_tr|MMB_ru|
> |-|-|-|-|-|-|-|-|
> |Qwen2-VL|Qwen2-7B|79.6|79.6|75.9|71.7|70.9|76.0|
> |LLaVA-OV|Qwen2-7B|77.1|76.6|73.2|66.9|65.5|71.3|
> |Parrot|Qwen2-7B|78.7|78.4|76.3|75.2|74.1|77.8|
>
> A4.2: For the radar chart in Figure 7, we have compared Parrot with the SOTA in 2024 (e.g., Mini-Gemini and LLaVA-Next), which shows Parrot’s competitiveness even with limited data. **Crucially, Parrot’s goal is not to surpass general-purpose MLLMs but to enhance multilingual ability with minimal data.**

---

### Official Review · Reviewer_852q · 2025-03-25

**Overall Recommendation:** 5

**Summary:**

The paper is addressing what authors call as "multi lingual erosion" in multimodal large language models (MLLMs) - a phenomenon where post multi modal alignment the model loses ability to respond in or process non-English inputs. The authors identify that existing vision-language alignment methods (LlaVa) use English centric data, thus resulting in visual features that are biased towards English and thus is generally poor towards other languages


To overcome the issue of multilingual erosion, the paper introduces Parrot, a novel multilingual visual instruction tuning method that uses textual guidance with Mixture of Experts to align visual features with multiple languages. Parrot, extracts image features via frozen encoder and then projects these into LLMs embedding space. Then, it performs a cross modal cross attention between visual features and token embeddings on input text. The cross attention output is fed to MoE router that activates language specific experts which then produces a language specific visual token representation by transforming English biased visual embeddings. Subsequently MoE re-weighting merges transformed features with original ones, there by retaining visual information. Thus image embeddings are first aligned with visual information prior to feeding them to the LLM.

Parrot is  then trained in a multi stage fashion - first, a modality alignment phase using a large English centric data keeping the vision encoder and the LLM frozen. The second stage is instruction tuning phase. for multilingual alignment where projector, MoE and LLM are tuned based on a multi-lingual instruction image text data set ~10k samples for each of the 5 language used in stage 2. These data sets were obtained using ShareGPT4V and GPT-4 translations with human calibrations. The base LLM used in Qwen-1.5 Chat which has strong multi-lingual capabilities.

Authors introduced Massive Multilingual Multimodal Benchmark (MMMB) to evaluate multilingual performance, comprising 12,000 visual QA instances spanning 6 languages (English, Chinese, Portuguese, Arabic, Turkish, Russian) across 15 categories​. They further extend the existing MMBench dev set to all six languages by translating questions via GPT-4 and manually verifying them​. Authors used a circular evaluation strategy where each multiple-choice question is converted into two yes/no questions to mitigate random guessing biases​. Beyond these multimodal ability is tested on a range of standard tasks.

The paper shows that this approach state-of-art (SOTA) on multilingual benchmarks, notably 14B obtains highest accuracy on all languages in MM Bench, and all non English languages on MMMB. 7B has a strong performance as well and exceed LLaVa-NeXT-13B. The gains have come without regressions to English performance. Similarly authors also show strong performance on general multimodal tasks performing competitively across tasks. The ablation studies demonstrate that MoE and the multilingual instruction tuning data set are critical to the performance of Parrot.

**Claims And Evidence:**

Overall the claims are generally well supported with convincing evidence.

1. Multilingual erosion: This is empirically well demonstrated using Chinese pre-trained CLIP yields better outputs (66.4 vs 68.3) on MM Bench-cn validation English biased encoder was an issue.

2. Parrot improving multi lingual alignment with little multilingual data: This is one of the strongest claims of this work and is well supported. Parrot is using ~1% data that competing models use (ex: Parrot 7B on 2M examples with ~10k per non-English language) outperforms Qwen-VL-Chat (1.1B English + 300M Chinese examples).  The ablation studies in table3 clearly establishes that using Parrot architecture adding small multilingual datasets incrementally can generate accuracy improvements

3. SOTA performance and improvements in underserved languages : Parrot's performance is SOTA across both benchmarks except English MMMB. This is well supported through comparisons in Table 2. Authors shows significant improvement on Turkish and Arabic exceeding prior SOTA by at least 10 points. These are substantial and adequately represented.

 Lastly authors show that the performance is maintained across multimodal tasks. This claim is supported by measuring on standard MM tasks - MME, MMStar etc. The data is represented in figure 5b and can be represented in a table for better clarity.

**Essential References Not Discussed:**

Paper is comprehensive addresses all major directions. No omission of essential work

Some recent notable references could be include - Pangea (Yue et al, 2024) is very recent work. Introduces fully open multi lingual MMLLM (39 languages) with instruction tuning dataset which 6M. Here authors show it significantly outperforms existing models in multi lingual settings. It is likely Pangea was published after the submission.

It would be interesting for authords to compare Parrot with Pangea, though Pangea has an alternate, complementary approach which is huge data and end-end training.

Similarly Maya - instruction tuned using a multilingualCLIP could be another reference to cite

Pali has been extended to PaliX with more capabilities, which is another reference to cite.

**Experimental Designs Or Analyses:**

The experimental design is comprehensive and sound.

1. Baselines - Parrot is compared a wide range of existing models open-source MLLMs (LLaVA 1.5, LLaVA-NeXT, Qwen-VL, mPLUG-Owl 2, InstructBLIP, MiniGPT-4v2 and closed source ones GPT-4V, Gemini, Qwen-VL-MAX. Authors used a unified eval framework which is VLMEvalKit. They have onboarded all models onto this eval framework which is a fair experimental evaluation approach.

2. MMMB and MM Bench:  The benchmarks seem comprehensive and robust. MMMB which was described in detail, consists of moderately difficult, multimodal questions across languages. They are not biased towards a specific language or task.
For MM Bench, GPT-4 was used for translation followed by human verification. Human verification followed by circular evaluation is  a strong design choice for evals which is fair and consistent

3. Ablations:  The ablations are well designed.
a) Multilingual data vs MoE-  (i) baseline (English-only), (ii) +multilingual data (but no MoE), (iii) +MoE (but maybe no multilingual data), and (iv) full Parrot. The authors show clear evidence that adding MoE improves the performance.
b) Incremental data ablation: Table 3 provides an analysis of how adding each language’s fine-tuning data affects performance​

There appear to be no flawed experiment design choices. The experiments are comprehensive.

**Methods And Evaluation Criteria:**

1. Methods are well motivated, and the approach looks sound. The idea of using language-specific experts to condition visual token on input language makes sense -  text conditioned transformation of visual-linguistic token in another language. Using MoEs and letting the experts choose specialized transformation seems appropriate. The modular approach, and two-stage training while not new are a reasonable strategy. Overall this methodology isn't significantly novel but extends current approaches to solve the multi lingual problem. There do not appear to be any fundamental flaws with this approach.

2. The MoE re-weighting is good design choice, that ensures visual features are constrained by original semantics. This is likely a contributor for Parrot's strong performance on generic tasks. However detailing on how the reweighting param was set and its impact would be useful to have.

3. Evals and benchmarks - This is a strong suite of the paper. They evals are thorough and appropriate. Specifically MMMB is well designed benchmark and fills prior gaps - wider array of languages and consistency. The authors also cover typologically different languages ensuring results are meaningful for a range of linguistic scenarios.

Beyond these authors evaluate on multimodal benchmarks to show that there  no notable regressions while comparing against an array of both open source and closed models. Using VLMEvalkit to evaluate all models ensures consistency and fairness.

 This is a strong, rigorous evaluation criteria.

**Other Comments Or Suggestions:**

Minor comments

Clarify Figure 2 Caption: The caption for Figure 2 is a bit unclear: “bad cases for multilingual benchmarkperceive”​

MoE Re-weighting Parameter: It’s mentioned that a trade-off parameter is used to blend original and expert outputs​. How is this chosen? Add more details to discuss this further.

**Other Strengths And Weaknesses:**

Strengths
- Strong Empirical Results and Data Efficiency - As stated previously significantly outperforms open models and most closed models with a fraction of data


- Originality of Approach: While built on existing components, the combination of cross-modal attention + MoE for language-specific visual token transformation is a novel idea in this space. To my understanding, no prior multimodal LLM work has used a mixture-of-experts to handle multiple languages dynamically. This is a fresh perspective compared to simply increasing training data or training separate models.   This is a non-trivial innovation that extends ideas from multilingual NLP into multimodal alignment.

Weaknesses
- Scalability to Many Languages: The paper doesn’t address how the method scales beyond the chosen languages. In principle, MoE could scale but there are challenges of requiring data in each language.  The authors do not test or discuss what happens if an unseen language is input to the model. There is strong dependency on Base LLM’s Language Ability:

**Questions For Authors:**

1. How is the MoE re-weighting parameter determined?

2. MoE Gating Mechanism: Do you use a hard gating or a soft combination of experts during inference?

3. Base LLM Choice and Multilingual Strength: You chose Qwen-1.5 as the LLM due to its strong multilingual capability. Have you tried Parrot with a base LLM that is less multilingual? How doed the base LLM impact performance?

4. Inference Efficiency: During inference, what is the computational overhead of Parrot compared to a standard pipeline (e.g., LLaVA)? Since you use a CLIP encoder (frozen) and an MoE, do you run the cross-attention and expert forward pass for every token, or just once per image?

4. High-Resolution Image Handling: In the limitations, you mention CLIP’s limitation with high-res or detail rich images. Have you considered straightforward mitigations like using CLIP-ViT-L/14 with a larger input size?

**Relation To Broader Scientific Literature:**

This work is well-positioned within existing literature - effectively builds on prior prior multilingual vision-language modeling, MoE applications, and multilingual instruction-tuned models. The key is the integration of MoEs with modular approaches to improve multilingual MMLLMs with minimal computational and data resources

- Multilingual Text Encoders for Vision: There have been attempts to retrofit models like CLIP to multilingual text. For example, mCLIP  learned a multilingual text encoder aligned with CLIP’s vision space via knowledge distillation​. Parrot’s approach is different in that it keeps the text input to the LLM multilingual, and instead adjusts the visual input.

- Large Multilingual Multimodal Models: The paper references PaLI (Chen et al., 2022)​  which is a 17B parameter model jointly trained on image-language data for 100+ languages.  Parrot’s contribution can be seen as a more data-efficient and parameter-efficient. Parrot adapts a 7B or 14B LLM with a small MoE module and achieves strong multilingual performance with a tiny fraction of data

- Instruction Tuning in Multilingual Context: The paper cites Ying-VLM-  instruction tuning an MLLM in English can indirectly generalize to other languages. Parrot’s work aligns with the philosophy that leveraging a multilingual LLM is key and goes further - explicitly tuning the visual features for each language, which yields far better results than hoping for zero-shot transfer. Parrot’s MoE module can be seen as bridging the gap between a multilingual instruction-tuned LLM and the monolingual vision features.

- Mixture-of-Experts in Multilingual Systems: There is relevant work in NLP that uses MoE for multilingual or multi-domain adaptation - Zhao et al. on language-guided MoE routing for machine translation. Parrot directly draws from this concept, as they mention in related work and implement a similar idea in the vision-language domain. Parrot’s novelty is applying it to align visual embeddings with multilingual text

**Theoretical Claims:**

There are  no theoretical claims that are put forth here.  This is primarily a empirical work.

---

> ### Author Rebuttal · Authors · 2025-04-01
>
> We are deeply grateful for the reviewer’s thorough and thoughtful assessment of our work, as well as their recognition of Parrot’s contributions.
>
> > **Q1: Scalability to many languages.**
>
> A1: Parrot’s MoE framework is inherently designed to support seamless integration of new languages. The addition of language-specific experts requires only minimal data (e.g., ~5k samples per language) and incurs a linear parameter overhead (one expert per language). Empirically, we observe robust knowledge transfer **within language families.** We will extend Parrot to encompass broader linguistic diversity in future work and present more case studies given an unseen language to the model.
>
> > **Q2: MoE re-weighting parameter.**
>
> A2: The trade-off parameter $\alpha$ in Eq. 5 balances the preservation of original visual semantics ($\alpha=0$) against language-specific transformation strength ($\alpha=1$). Through grid search on MMMB validation data with $\alpha \in \{0.1, 0.3, 0.5, 0.7, 1.0, 1.5\}$, we found $\alpha=0.5$ optimally maintains visual fidelity while enabling robust multilingual alignment (>5% gain over $\alpha=0.1$), as excessive reliance on original English-biased features hindered language adaptation. Higher $\alpha$ values (>1.0) caused degradation in general visual tasks (e.g., -2.2% on MMBench object counting). We will include full parameter sensitivity curves in the final version to clarify this design choice.
>
> > **Q3: MoE gating mechanism.**
>
> A3: As described in Eq. 4, we use soft gating (weighted combination of experts) rather than hard routing during inference. This ensures smooth transitions between languages, avoids overfitting to dominant experts, and dynamically handles multilingual prompts (e.g., mixed-language queries). We will add a discussion in §3.3 highlighting this design choice.
>
> > **Q4: Base LLM impact.**
>
> A4: We ablated this by replacing Qwen1.5-7B with Vicuna1.5-7B (weaker multilingual support). As shown in the table below, Qwen1.5 outperforms Vicuna1.5 by +4.4% on Turkish and +8.5% on Arabic. This highlights that Parrot’s effectiveness is somewhat reliant on the base LLM’s multilingual ability. **Notably, while using the weaker multilingual LLM, the performance gain compared to the baseline LLaVA-1.5 is remarkably large.**
>
> |Method|LLM|MMMB_en|MMMB_zh|MMMB_pt|MMMB_ar|MMMB_tr|MMMB_ru|
> |-|-|-|-|-|-|-|-|
> |LLaVA-1.5|Vicuna1.5-7B|67.1|58.8|59.8|43.5|46.4|59.1|
> |Parrot|Vicuna1.5-7B|68.2|65.4|64.3|54.2|53.6|63.0|
> |Parrot|Qwen1.5-7B|70.0|68.1|67.3|62.7|58.0|66.3|
>
> > **Q5: Inference efficiency.**
>
> A5: Parrot maintains inference efficiency comparable to LLaVA by executing cross-attention and MoE processing **once per image** during visual token projection, not per generated token. The lightweight MoE module adds <0.5% parameters to the LLM, resulting in almost no extra runtime latency increase versus LLaVA-1.5 under identical hardware. Frozen CLIP encoding and single-pass visual-language alignment ensure the computational overhead remains negligible despite enhanced multilingual capabilities.
>
> > **Q6: High-resolution image handling.**
>
> A6: Thank you for your valuable suggestion. Parrot currently uses CLIP ViT-L/14-336 that is the maximum input size version. Crucially, our current implementation prioritizes fair comparison with LLaVA-1.5 baselines that share the same CLIP backbone. In future work, we plan to integrate dynamic-resolution approaches like NaViT's flexible patching or SigLIP's high-res pretraining to better handle high-resolution images while maintaining multilingual alignment capabilities.
>
> > **Q7: Minor revisions.**
>
> A7: Thanks for the detailed review and helpful comments. We will incorporate these updates into the final version.
> 1) **Figure 2 caption**: We have clarified the caption to "Examples of suboptimal multilingual benchmark design in existing works, as evaluated against MMMB's principles."
> 2) **Reference**: we shall include a comprehensive comparative analysis of multilingual MLLM's recent advancements in the final version.

---

### Decision · Program_Chairs · 2025-05-01

**Decision:**

Accept (poster)

**Comment:**

This paper addresses the issue of "multilingual erosion", where MLLM loose proficiency in non-English languages after multimodal alignment training (e.g. answering in English while asking non-English questions). PARROT presents a novel approach to enhance the multilingual capabilities of MLLMs, using language specific embeddings to fuse to visual embeddings and multilingual MoEs. To be able to  evaluate multilinguality, the paper also introduces a new Massive Multilingual Multimodal Benchmark (MMMB). Experiments show SOTA results on MMMB, MMBench and other multimodal tasks. During rebuttals, authors clarified and resolved reviewers' concerns. While this is primarily empirical work, the paper solves an important problem and makes a substantial contribution to the field of multi-modal LLMs.